# Natural variability in bee brain size and symmetry revealed by micro-CT imaging and deep learning

Philipp D. Lösel [1,2,3‡]*, Coline Monchanin [4,5‡]*, Renaud Lebrun [6,7], Alejandra Jayme [1,2], Jacob J. Relle [1,2], Jean-Marc Devaud [4], Vincent Heuveline [1,2,8‡], Mathieu Lihoreau [4‡]

1 Engineering Mathematics and Computing Lab (EMCL), Interdisciplinary Center for Scientific Computing (IWR), Heidelberg University, Heidelberg, Germany, 2 Data Mining and Uncertainty Quantification (DMQ), Heidelberg Institute for Theoretical Studies (HITS), Heidelberg, Germany, 3 Department of Materials Physics, Research School of Physics, The Australian National University, Canberra, Australia, 4 Research Center on Animal Cognition (CRCA), Center for Integrative Biology (CBI); CNRS, University Paul Sabatier – Toulouse III, Toulouse, France, 5 Department of Biological Sciences, Macquarie University, Sydney, Australia, 6 Institut des Sciences de l'Evolution de Montpellier, CC64, Université de Montpellier, Montpellier, France, 7 BioCampus, Montpellier Ressources Imagerie, CNRS, INSERM, Université de Montpellier, Montpellier, France, 8 Heidelberg University Computing Centre (URZ), Heidelberg, Germany

‡ PDL and CM are joint first authors on this work. VH and ML are joint senior authors on this work.
* philipp.loesel@anu.edu.au (PDL); coline.monchanin@gmail.com (CM)

**Data Availability Statement:** Three-dimensional image and label data of honey bees and bumblebees as well as trained networks are available at https://biomedisa.org/gallery/. The

## Abstract

Analysing large numbers of brain samples can reveal minor, but statistically and biologically relevant variations in brain morphology that provide critical insights into animal behaviour, ecology and evolution. So far, however, such analyses have required extensive manual effort, which considerably limits the scope for comparative research. Here we used micro-CT imaging and deep learning to perform automated analyses of 3D image data from 187 honey bee and bumblebee brains. We revealed strong inter-individual variations in total brain size that are consistent across colonies and species, and may underpin behavioural variability central to complex social organisations. In addition, the bumblebee dataset showed a significant level of lateralization in optic and antennal lobes, providing a potential explanation for reported variations in visual and olfactory learning. Our fast, robust and user-friendly approach holds considerable promises for carrying out large-scale quantitative neuroanatomical comparisons across a wider range of animals. Ultimately, this will help address fundamental unresolved questions related to the evolution of animal brains and cognition.

## Author summary

Bees, despite their small brains, possess a rich behavioural repertoire and show significant variations among individuals. In social bees this variability is key to the division of labour that maintains their complex social organizations and has been linked to the maturation of specific brain areas as a result of development and foraging experience. This makes

source code is available as part of the Biomedisa open source project. It was developed and tested for Ubuntu 22.04 LTS and Windows 10. Biomedisa can be used via the command line or with any common browser as an interface. The source code can be downloaded from the official GitHub repository at https://github.com/biomedisa/biomedisa/. To install the software, please follow the provided installation instructions. Specifically, the evaluation and R script, along with the volumetric measurements, required for replication and analysis are also available in this GitHub repository. All other relevant data are included within the paper and its Supporting Information files.

**Funding:** Three-dimensional data acquisitions were performed using the micro-CT facilities of the MRI platform member of the national infrastructure France-BioImaging supported by the French National Research Agency (ANR-10-INBS-04, «Investments for the future»), and of the Labex CEMEB (ANR-10-LABX-0004), and NUMEV (ANR-10-LABX-0020), which supported RL. We further acknowledge the support by the projects ASTOR (05K2013) and NOVA (05K2016) funded by the German Federal Ministry of Education and Research (BMBF), Informatics for Life funded by the Klaus Tschira Foundation, the state of Baden-Württemberg through bwHPC, the Ministry of Science, Research and the Arts Baden-Württemberg (MWK) through the data storage service SDS@hd, and the German Research Foundation (DFG; INST 35/1314-1 FUGG and INST 35/1134-1 FUGG), which supported PDL, AJ, JJR, and VH. CM was funded by a PhD fellowship from the French Ministry of Higher Education, Research and Innovation. JMD and ML were funded by the Agence Nationale de la Recherche (3DNaviBee ANR-19-CE37-0024), the Agence de la Transition Ecologique (project LOTAPIS), and the European Commission (FEDER ECONECT MP0021763, ERC Cog BEE-MOVE GA101002644). The funders had no role in study design, data collection and analysis, decision to publish, or preparation of the manuscript.

**Competing interests:** The authors have declared that no competing interests exist.

bees an ideal model for understanding insect cognitive functions and the neural mechanisms that underlie them. However, due to the scarcity of comparative data, the relationship between brain neuro-architecture and behavioural variance remains unclear. To address this problem, we developed an AI-based approach for automated analysis of three-dimensional brain images and analysed an unprecedentedly large dataset of honey bee and bumblebee brains. Through this process, we were able to identify previously undescribed anatomical features that correlate with known behaviours, supporting recent evidence of lateralized behaviour in foraging and pollination. Our method is open source, easily accessible online, user-friendly, fast, accurate, and robust to different species, enabling large-scale comparative analyses across the animal kingdom. This includes investigating the impact of external stressors such as environmental pollution and climate change on cognitive development, helping us understand the mechanisms underlying the cognitive abilities of animals and the implications for their survival and adaptation.

## Introduction

Artificial intelligence is helping scientists to more efficiently and effectively analyse data in a wide range of scientific fields, enabling them to make new discoveries and address important open questions [1]. In particular, neuroscience is a field that can greatly benefit from automated analysis tools for large-scale comparative investigations. Animals, from insects to humans, show a rich diversity of behavioural profiles, or personalities, that may be underpinned by anatomical and cognitive variability [2]. Identifying natural variations in brain morphology and performance at the intra- and inter-specific levels can therefore help to understand the evolution of species and their potential for resilience to environmental stressors in the context of biodiversity loss [3]. However, such an approach currently requires extensive manual effort in order to obtain and analyse large and high-quality brain datasets, limiting investigations to a few individuals and model species for which brain atlases are available (e.g. rodents [4], primates [5], *Drosophila* [6]), thereby restraining the scope for comparative research [7].

Three-dimensional (3D) imaging techniques, such as micro-computed tomography (micro-CT), offer the potential to facilitate large-scale investigations. These methods enable non-destructive, fine-scale imaging of internal structures of biological objects, including brain tissues, both *in vivo* and *ex vivo*. Recent improvements in resolution and acquisition times [8,9] have broadened the application of micro-CT to a wide range of animals, including various species of small-sized insects (e.g. ants [10], wasps [11], beetles [12], bees [13,14]), and posed new demands for accelerated image analysis methods. However, image segmentation and post-processing steps still require manual assistance [15–18], which is time-consuming (several hours or days to reconstruct a structure) and not feasible for analysing large datasets.

Here we demonstrate how deep learning can significantly speed up 3D image analysis for large-scale comparative brain studies in insects by using the semi-automated [19,20] and automated segmentation methods of the recently developed online platform Biomedisa [21] (https://biomedisa.org) to analyse 3D micro-CT image data from bee brains (Fig 1).

Despite their small size ($<1mm^3$), bee brains support a rich behavioural repertoire and exhibit extensive inter-individual variability [22]. In social bees, such as honey bees and bumblebees, this variability is central to the division of labour that supports complex social organisations, and some of it has been linked to the maturation of specific brain areas due to development and foraging experience [23,24]. This makes bees ideal models for studying

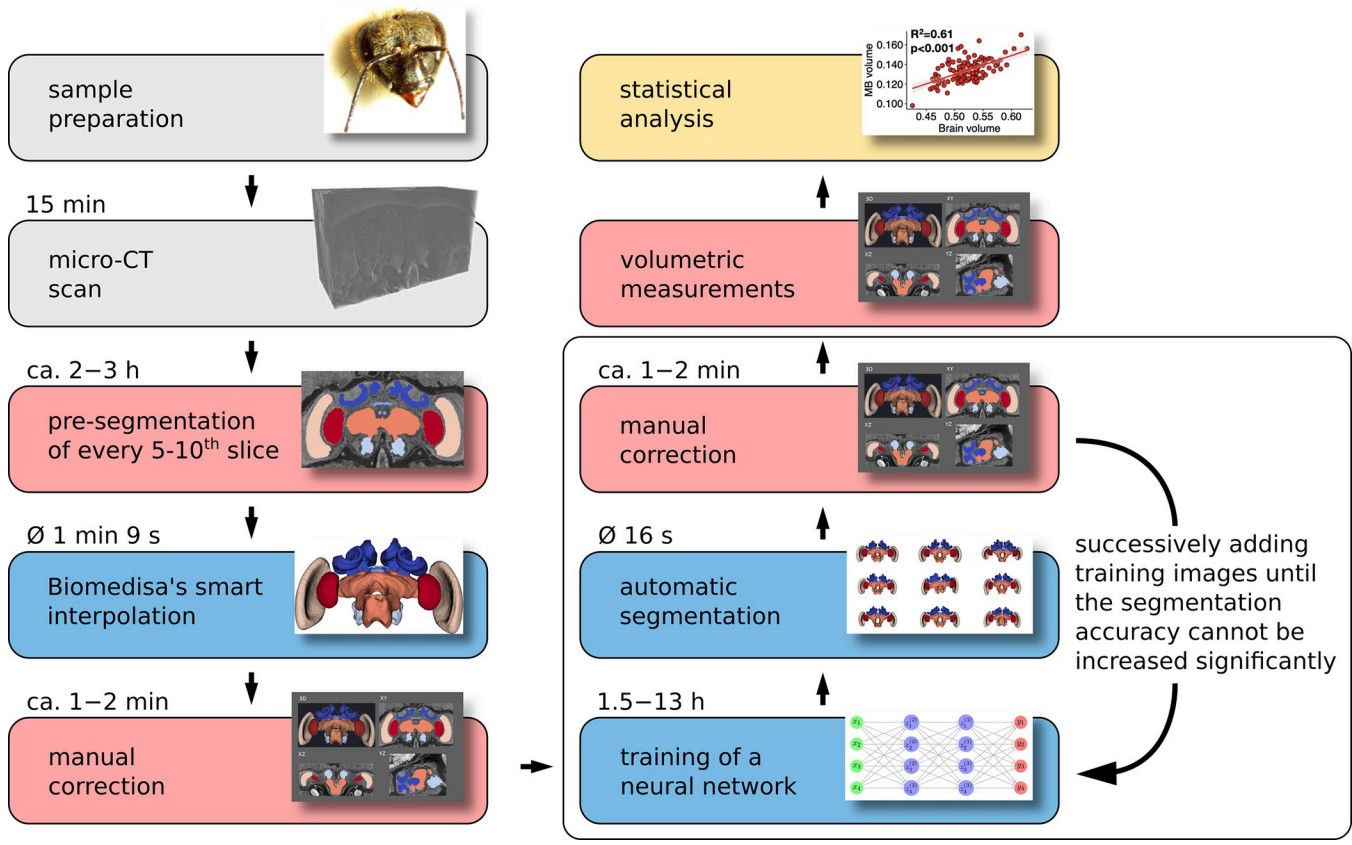

**Fig 1. Flowchart of the steps to perform large-scale quantitative comparative analyses of bee brain size and organisation using micro-CT imaging and the Biomedisa segmentation platform.** After sample preparation and volume reconstruction (*grey* boxes), the micro-CT scans are segmented with AVIZO 2019.1 (*red* boxes) in combination with Biomedisa (*blue* boxes). The volumes are then measured with AVIZO 2019.1 and statistically analysed with R Studio (*yellow* box). For the honey bee dataset, the loop within the white box represents the training process for 3, 7, 12, 18, and 26 brains. Each step involved a different number of segmented and manually corrected 3D images, specifically 4, 5, 6, and 8 respectively. Upon completing the loop, the remaining 84 brains were segmented and manually corrected. All processing times provided are average times per brain.

insect cognitive functions and their underlying neural substrates [25,26]. However, there is currently no clear connection between possible differences in brain neuro-architecture and behavioural variance due to the lack of data from comparative analyses (few studies [27] and sample sizes ranging from 7 [28] to 67 [29] individuals - [29] used 67 European honey bees and 54 Africanised honey bees).

By collecting 3D image data from the brains of 110 honey bees (*Apis mellifera*) and 77 bumblebees (*Bombus terrestris*) and using semi-automated segmentation to annotate the training data for a deep neural network, we achieved a substantial reduction in data processing time, up to 98%, when compared to a conventional manual segmentation approach. Our results revealed size variations and strong anatomical asymmetries in the brain samples, supporting behavioural observations from the literature. In addition to an extensive evaluation and a detailed protocol of Biomedisa's deep learning method, and the new insights into the evolution of bee brains, this is the first study to our knowledge that will publish an unprecedentedly large dataset of the underlying 3D image and label data of bee brains. Our method is open access, readily available online, fast, accurate, robust to different species, and can be applied to a wide range of 3D imaging modalities and scientific questions beyond comparative neurosciences.

## Results and discussion

### Automatic segmentation of bee brains considerably speeds up analysis

We performed micro-CT scanning of 120 honey bee foragers (*A. mellifera*, Buckfast) from two apiaries (Population A: 100 bees from 6 hives, population B: 20 bees from 3 hives) in Toulouse (France). Brain samples were prepared following Smith et al. [14] and CT-scanned at a resolution of 5.4 μm isotropic voxel size (see "Methods"). Among the 120 micro-CT scanned brains, 10 were damaged during manipulation and discarded, leaving 110 brains for our analysis (S1 Appendix). Image dimensions and image spacing varied across subjects and averaged 844×726×485 isotropic voxels and 0.0054×0.0054×0.0054 mm³, respectively.

We analysed six major brain neuropils based on the 3D bee brain atlas [30]: the antennal lobes (AL) that receive olfactory information from the antennae; the mushroom bodies (MB, each comprising the medial and lateral calyces, peduncle and lobe) that integrate olfactory and visual information; the central complex (CX, comprising the central body, the paired noduli and the protocerebral bridge) that receives compass and optic flow information; the medullae (ME) and lobulae (LO) that receive visual information from the compound eyes, combined together as "optic lobes" (OL) in our analysis (retinae and laminae were not measured); and the other remaining neuropils (OTH) (including protocerebral lobes and subesophageal zone) (Fig 2). ALs and OLs are involved in olfactory and visual processing, respectively, while the CX and MBs play important roles in locomotor behaviour, learning and memory, respectively [31]. Total brain volume was calculated as the sum of AL, MB, ME, LO, CX, and OTH.

The volumetric analysis of the different neuropils required their isolation from the 3D image data by segmentation (Figs 1 and 2). Each dataset was manually cropped to the area of the neuropils (Fig 2D) using AVIZO 2019.1, resulting in an average size of 451×273×167

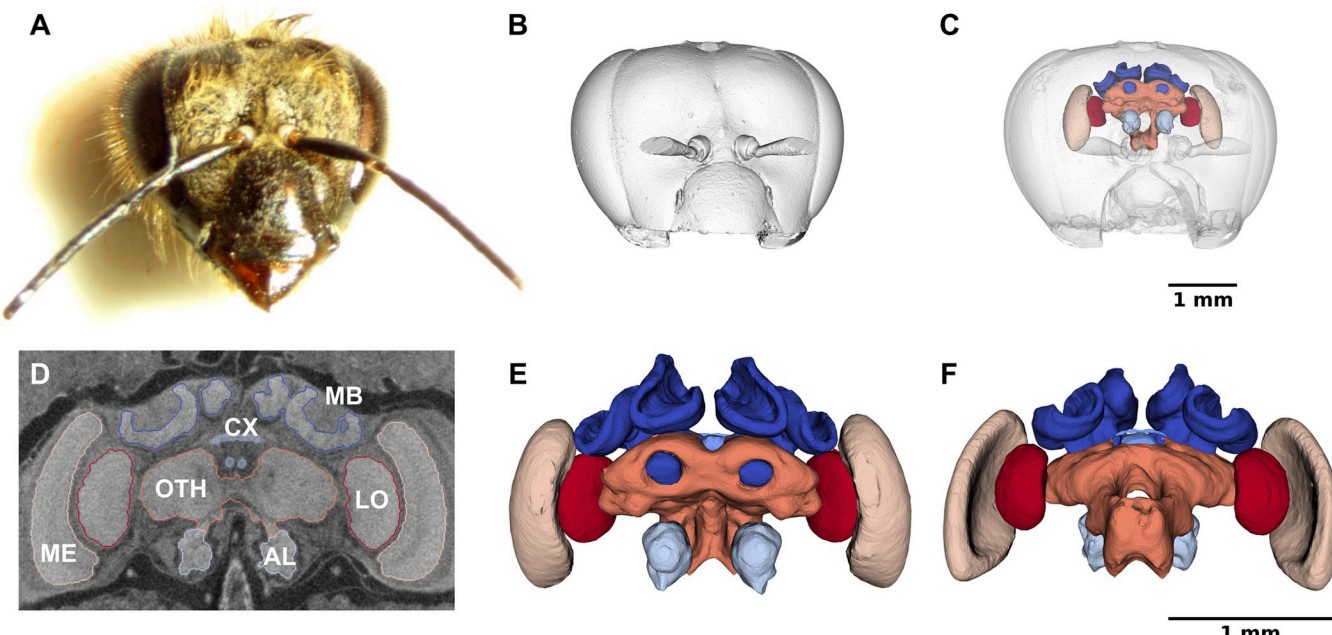

**Fig 2. Surface renderings of an example of CT-scanned honey bee head and reconstructed brain neuropils.** (**A**) Frontal view of the head of a forager bee (ID 79, hive H4). (**B**) Surface rendering of the head with the mandibles removed. (**C**) Overlay of the head and reconstructed neuropils. (**D**) Frontal cross-section of the tomogram with the segmentation boundaries of the mushroom bodies (MB), central complex (CX), antennal lobes (AL), medullae (ME), lobulae (LO) and other neuropils (OTH). (**E**) Frontal view of the reconstructed MB (*dark blue*), CX (*sky blue*), AL (*light sky blue*), ME (*beige*), LO (*red*) and OTH (*orange*). (**F**) Dorsal view of the reconstructed neuropils. (B), (C), (E) and (F) were created with ParaView Glance integrated in Biomedisa.

voxels. For automatic segmentation, we used the deep neural network interface from the online platform Biomedisa. To train a deep neural network, Biomedisa needs a set of fully segmented 3D images. To create the first three datasets, labels were assigned manually to the six neuropils in every 10th slice and every 5th slice in the interval containing CX within the 3D volume using AVIZO 2019.1 (Thermo Fisher Scientific, Waltham, USA). For better performance compared to conventional morphological interpolation (see "Methods" and Fig 3), Biomedisa's semi-automatic smart interpolation was used to segment the remaining volume between the pre-segmented slices. Before interpolation, the 3D image data was slightly smoothed using Biomedisa's "denoise" function (see "Methods"). Subsequently, outliers (i.e. unconnected voxels or islands) were removed and segmentation errors were corrected manually by an expert using AVIZO 2019.1.

Additional training data was then iteratively created using neural networks. Starting from three semi-automatically segmented 3D images, we trained neural networks on 3, 7, 12, 18, and 26 three-dimensional images by adding manually corrected segmentation results of the last trained network to the training data after each step (Fig 1). In all cases, the network's default configuration was used (see "Methods"). The corresponding training times on 4 NVIDIA Tesla V100s were 1.5, 3.5, 6, 9, and 13 hours. At this point, segmentation accuracy could no longer be significantly improved by additional training data (Fig 3 and S1 Table). We finally used the network trained on the set of 26 three-dimensional images ("3D training images") to automatically segment the remaining 84 micro-CT scans of honey bee brains ("3D test images"). The automatic segmentation of each 3D image took an average of 16 seconds using an NVIDIA RTX A5000 or 82 seconds using an Intel Core i9-11900K Processor. All segmentation results were checked and manually corrected by an expert (using "remove islands" and "merge neighbours" functions in AVIZO 2019.1).

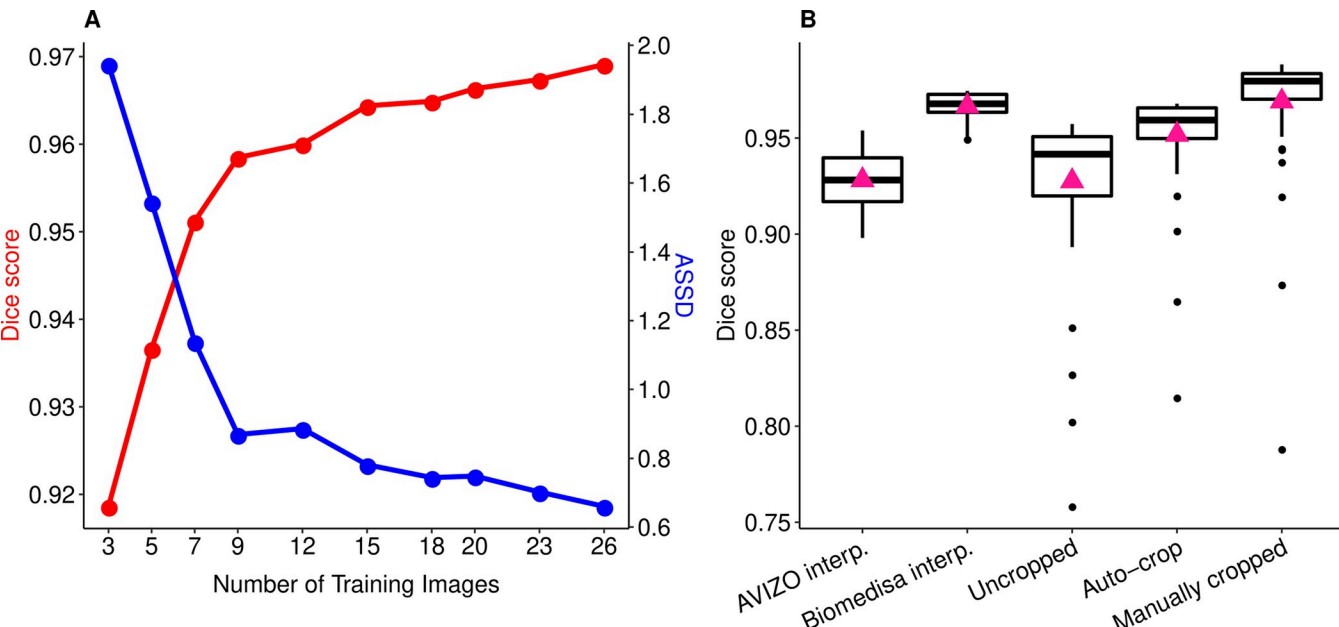

**Fig 3. Segmentation accuracy of Biomedisa's semi-automatic and automatic segmentation of honey bee CT scans.** (**A**) Average Dice scores (*red*) and average symmetric surface distances (ASSD, *blue*) of the automatic segmentation results for an increasing number of 3D training images. (**B**) Semi-automated segmentation accuracy (Dice score) of the AVIZO interpolation and the Biomedisa interpolation as well as segmentation accuracy of the automatic segmentation for uncropped 3D image data, using Biomedisa's auto-cropping and manually cropped 3D image data. Boxplots show median volumes (intermediate line) and quartiles (upper and lower lines). Pink triangles display mean volumes. For performance tests of the automatic segmentation, the 84 three-dimensional honey bee test images were split into 30 three-dimensional validation images and 54 three-dimensional test images (see "Methods").

To test the suitability of our approach for brain datasets from other species (that, unlike honey bees [26], are not well-established model species in neurosciences), we also performed micro-CT scans of 77 bumblebees (*B. terrestris*) from 4 commercially-available colonies (colony A: 18 bumblebees, colony B: 20 bumblebees, colony C: 21 bumblebees, colony D: 18 bumblebees) (S1 Appendix). Based on the evaluation for an increasing number of 3D training images (see "Methods", Fig 3 and S1 Table), we decided to semi-automatically segment 13 three-dimensional training images (which we considered to be a good balance between manual effort and accuracy) using Biomedisa to train a neural network on the bumblebee dataset. The trained network was then used to automatically segment the remaining 64 CT scans.

Overall, AVIZO 2019.1 was used for pre-segmentation, correction of segmentation results and measurement of absolute neuropil volumes calculated with the voxel count function. Biomedisa was used for smart interpolation to create the initial training data, training of deep neural networks and subsequent automatic segmentation. Since Biomedisa supports the AMIRA Mesh File format, deployed by AVIZO and AMIRA, data can be easily transferred between the Biomedisa online platform and AVIZO 2019.1. This integration ensures the preservation of label values and names, streamlining the exchange of data and facilitating seamless collaboration between platforms. In addition to AMIRA meshes (AM), Biomedisa supports many other common data formats (such as DICOM, NifTI, TIFF, NRRD, MHA, MHD) and can also be used in combination with other segmentation tools.

For both datasets, this automatic segmentation considerably reduced the time and effort required for 3D image analysis compared to conventional manual segmentation (manually segmenting individual slices, followed by linear interpolation and manual correction). For each 3D image, it took about 5 to 10 minutes for the whole procedure: importing the data into AVIZO 2019.1, cropping to the area of the neuropils, exporting the data, uploading it to Biomedisa, performing automatic segmentation, downloading and importing the segmentation result into AVIZO 2019.1, correcting the segmentation result manually, and measuring volumes. In contrast, a conventional manual or semi-automatic approach would have taken several hours for each CT scan (approx. 5 to 6 h using conventional manual segmentation and 2 to 3 h using Biomedisa's semi-automatic segmentation). Depending on the quality of the automated segmentation result, either no correction was required (S1A Fig) or manual correction typically took 1 to 2 minutes (S1B Fig), rarely longer if the result was significantly flawed (S1C Fig). Typical artefacts resulting from the automatic segmentation are outliers (S1B Fig) which can be easily removed either with Biomedisa's cleaning function or with AVIZO 2019.1. In the end, with our approach, the manual effort for segmenting the full honey bee dataset (15 to 27 h) was about 95 to 98% less than conventional manual segmentation (550 to 660 h) and about 87 to 93% less for the bumblebee dataset (31 to 50 h compared to 385 to 462 h) (see "Methods").

## Accurate automatic segmentation of bee brains with noticeable performance variations in bumblebees

We evaluated the accuracy of the automatic segmentation using two complementary and commonly applied metrics for measuring performance in biomedical image segmentation challenges [32] (see "Methods"): the Dice similarity coefficient (Dice) and the average symmetric surface distance (ASSD). The Dice score quantifies the matching level between two segmentations and is between 0 (no overlap at all) and 1 (perfect match). The ASSD is the average 3D Euclidean distance from a point on the surface of one segmentation to the closest point on the surface of the second segmentation and vice versa. The smaller the ASSD, the closer the two segmentations.

For both bee species, we measured the accuracy of the segmentation results of the 3D test images performed by the deep neural network trained with the respective training data. We compared the results obtained without further manual post-processing with ground truth data generated by an expert (i.e. segmentation results after revision and manual correction). For the automatic segmentation of the 84 honey bee brains, a Dice score of 0.987 was achieved (S1 Table). In 10.7% of the 3D test images, little or no manual correction was necessary (error less than 0.01% of Dice score, S1A Fig). In 84.5%, a slight manual correction was required taking 1 to 2 minutes (error ranging between 0.01% and 4%, S1B Fig). Only 4.8% of the segmentation results were significantly flawed (error greater than 4%, S1C Fig), usually due to a significant deviation from the training data (e.g. 3D images from incompletely stained or damaged brains during tissue processing) and required extensive manual correction or semi-automatic reconstruction using Biomedisa's smart interpolation. For the automatic segmentation of the 64 bumblebee brains, a Dice score of 0.983 was achieved (S1 Table). Here, no error was less than 0.01%. However, for 9.4% of the segmentation results, the error was less than 0.1% (S2A Fig). For 78.1% it was between 0.1% and 4% (S2B Fig) and 12.5% had an error greater than 4% (S2C Fig).

Biomedisa thus enabled accurate and fast segmentation of the 148 three-dimensional test images (84 honey bees, 64 bumblebees), which required manual correction of the segmentation results equalling a Dice score of 0.013 (i.e. an error of 1.3%) for the honey bee brains and 0.017 for the bumblebee brains (S1 Table). Major corrections were mostly only required for the smallest brain area considered, the CX (an average error of 3.4% for honey bees and 16.2% for bumblebees). Its fine structure combined with low contrast often makes it difficult to detect this neuropil, particularly in the bumblebee CT scans (S2 Fig). Overall, the segmentation exhibited greater variability in bumblebees, with a notably higher number of samples displaying errors exceeding 4%. However, it is crucial to emphasise that the relatively lower performance in bumblebees is primarily attributed to the lower image quality of the bumblebee data (S2 Fig), rather than being a consequence of overfitting caused by the smaller number of 3D training images used (13 in bumblebees compared to 26 in honey bees). In fact, fine-tuning the network initially trained on the honey bee dataset specifically for the bumblebee dataset (with a fixed decoder part and only training the encoder part [33]) yielded improved results only when the number of 3D training images was limited (3 to 7 three-dimensional training images), while the advantage diminished when more than 12 three-dimensional training images were employed (S3 Fig). Additionally, similar to the evaluation with an increasing number of 3D training images in honey bees (see "Methods", Fig 3 and S1 Table), the rate of improvement observed when increasing the number of 3D training images in bumblebees slowed down when utilising more than 12 three-dimensional training images (S3 Fig).

The results presented were achieved using Biomedisa's standard configuration (see "Methods"). Altering parameters such as network size, batch size, or learning rate did not lead to improved segmentation accuracy but often resulted in notably inferior outcomes (S2 Table). However, there was no significant difference in segmentation accuracy between a maximum of 512 channels and the maximum of 1024 channels but reduced computation time from 13 to 10 hours. Additionally, while both the standard accuracy and Dice score remain close to their maximum values up to 200 epochs (S4–S7 Figs), indicating minimal risk of overfitting, the loss function exhibited a slight increase starting around 100 epochs (S5 and S7 Figs). Consequently, reducing the network size and number of epochs may enhance (energy) efficiency without compromising segmentation accuracy. The decision to use 200 epochs and 1024 channels was made to ensure thorough training of the network.

## Brain volumes varied significantly among honey bees

To validate our segmentation approach, we compared our results with previously published data, focusing on the volumes of the six neuropils considered (Fig 4) in honey bees. Division of labour in honey bee colonies is primarily based on age differences among individuals [34]. Since workers vary little in body size [35], it was expected that their brain volume would also show only small variation. Our results are consistent with previous studies based on smaller datasets [24,28–30,36–42] and using different quantitative approaches (CT scan data [41,42], or stereo [24,28,29,38], confocal [30,36,37], and nuclear magnetic resonance [36] microscopy). When data do not match, we expect our measures to be more accurate than those in previous studies since our measurements were taken from thinner slices (5.4 μm vs. 8 [30] to 60 [39]

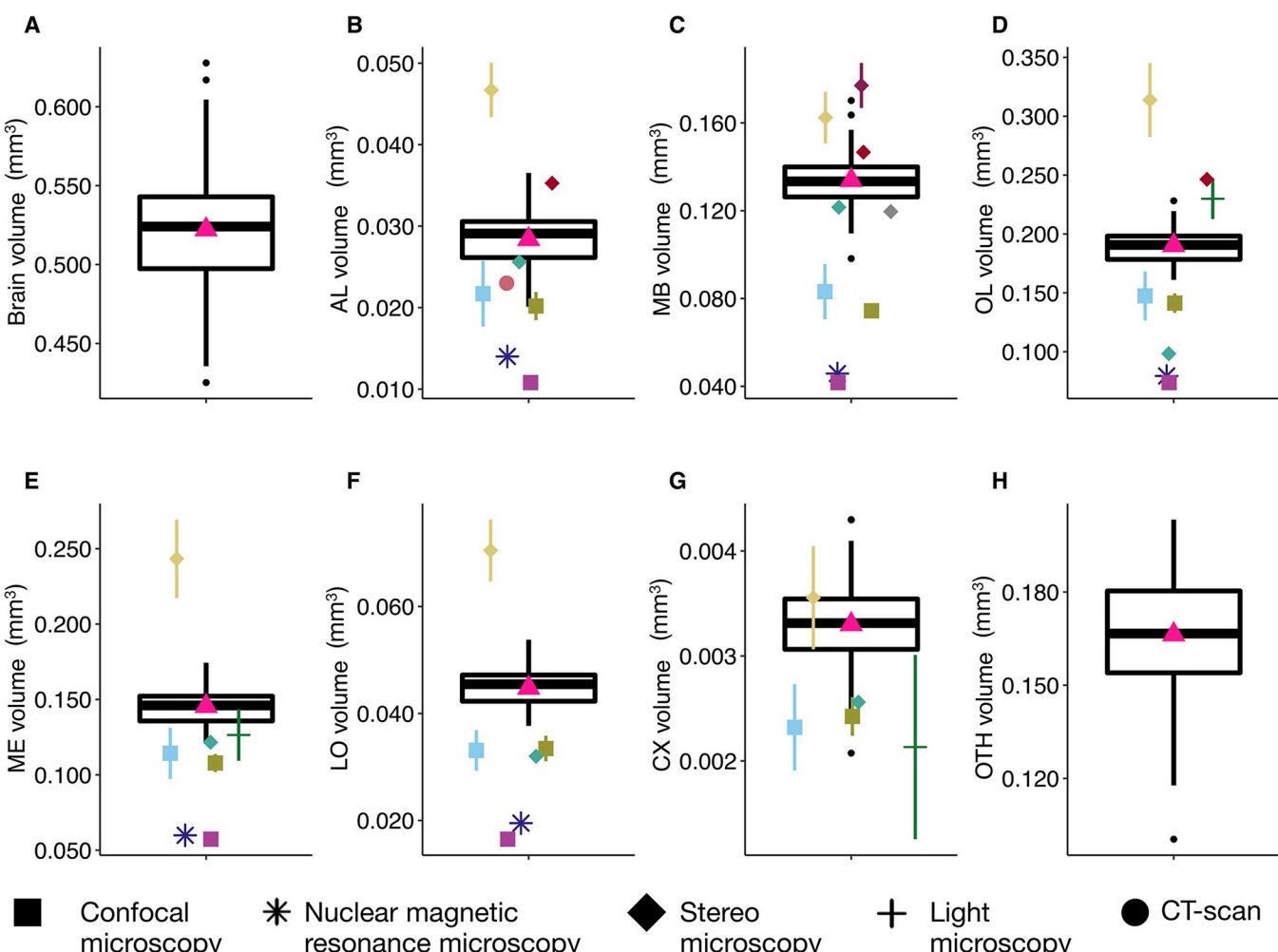

**Fig 4. Variation in brain and neuropils volumes (mm³) for honey bees ($N$ = 110).** (**A**) Total brain. (**B**) Antennal lobes (AL). (**C**) Mushroom bodies (MB). (**D**) Optic lobes (OL). (**E**) Medullae (ME). (**F**) Lobulae (LO). (**G**) Central complex (CX). (**H**) Other neuropils (OTH). Boxplots show median volumes (intermediate line) and quartiles (upper and lower lines). Pink triangles display mean volumes. Coloured symbols show mean (± s.d. when available) of neuropil volumes described for forager honey bees in other studies: using confocal microscopy (*square*): Brandt et al. [30] ($N$ = 20 bees - *light blue*); Steijven et al. [37] ($N$ = 10 - *kaki*); Haddad et al. [36] (*purple*); using nuclear magnetic resonance microscopy (*star*): Haddad et al. [36] ($N$ = 8 - *deep blue*); using stereo microscopy (*diamond*): Gowda & Gronenberg [28] ($N$ = 7 - *yellow*); Mares et al. [38] ($N$ = 25 - *turquoise*); Maleszka et al. [39] ($N$ = 30 - *burgundy*); Withers et al. [24] (*red*); Durst et al. [40] ($N$ = 12 - *grey*); using light microscopy (*cross*): Gronenberg & Couvillon [29] ($N$ = 121 European and Africanized honey bees - *dark green*); using CT scan (*point*): Greco and Stait-Gardner [41] ($N$ = 10 - *coral*). For the total brain volume, comparisons with other studies are not shown because some studies did not provide total brain volume information, while others measured different neuropils compared to our study.

μm) and on all successive slices (rather than leaving intermediate slices aside and using linear interpolation, e.g. Cavalieri principle [43]). As histological procedures differ across studies, we cannot exclude that variations in the extent of tissue shrinkage may also contribute to differences across datasets. In addition, CT imaging does not always allow a clear separation between connective tracts and neuropiles, which may lead to a slight over-estimation of some neuropil volumes (without any expected impact on inter-individual comparisons). Such technical discrepancies likely explain under- or over-estimated volumes in other studies compared to ours. Also note that some measurements from other studies have uncertainties, e.g. a precise description of which structures were included in the measurement of CX [30] or whether or not cell bodies were included in the total brain volume [28]. Additionally, differences in the estimated structural volumes may arise from biological differences among samples taken from bees of different subspecies (e.g. European and Africanized honey bees [29]), ages and foraging experiences of individual bees [24,40].

Among our 110 honey bee brains, total brain volume (i.e. sum of all measured neuropils) varied by 32% and neuropil volumes by 29 to 52% (Fig 4 and Table 1). We found strong positive correlations between the absolute volumes of all neuropils - but the CX - and total brain volume (Fig 5A–5G). Most neuropils scaled isometrically with total brain volume (Fig 5H), with only a lower correlation coefficient for CX ($r = 0.36$, $p<0.001$; Fig 5F). This is coherent with the results of the largest comparative analysis of honey bee brains so far (121 brain samples obtained by microscopy technique in two honey bee strains) [29]. When considering relative volumes (ratio of neuropil volume to total brain volume) (Table 1), we found only a positive correlation between OTH and total brain volumes, while relative volumes of MB, OL, ME and CX correlated negatively with total brain volume (S8 Fig).

When assessing correlations between absolute volumes across neuropils, we found strong correlations between OL, LO and ME volumes (S3 Table), as previously reported [29]. In our

**Table 1. Total brain and neuropil volumes of honey bees ($N = 110$).** Mean (± standard deviation), minimal and maximal volumes (mm³), percentage of volume variation ((Max−Min)×100/Max) and relative volume (%). For paired neuropils, detailed data for both sides (left/right) are given. Note that the left/right comparison for MB is based on data from 59 honey bees only, as some bees have both sides merged in the automatic segmentation result. F-test, following LMMs, tests the significance of the fixed variable "hive", and results are displayed in bold when significant.

| Neuropil | Mean ± s.d. (mm³) | Min (mm³) | Max (mm³) | % variation | F-test | Relative volume ± s.d. (%) |
|---|---|---|---|---|---|---|
| **Brain** | 0.522±0.038 | 0.425 | 0.628 | 32.27% | $F(8,101) = 1.018$, $p = 0.428$ | |
| **AL** | 0.0284±0.0033 | 0.0201 | 0.0365 | 44.97% | **$F(8,101) = 5.055$, $p<0.001$** | 5.43±0.47 |
| *Left* | *0.0142±0.0017* | *0.0095* | *0.0183* | *47.76%* | | |
| *Right* | *0.0141±0.0018* | *0.0096* | *0.0182* | *47.49%* | | |
| **MB** | 0.134±0.011 | 0.098 | 0.170 | 42.30% | **$F(8,101) = 2.315$, $p = 0.025$** | 25.68±1.75 |
| *Left* | *0.066±0.007* | *0.030* | *0.075* | *60.25%* | | |
| *Right* | *0.065±0.005* | *0.048* | *0.073* | *34.26%* | | |
| **OL (=ME+LO)** | 0.190±0.014 | 0.161 | 0.228 | 29.41% | $F(8,101) = 4.368$, $p = 0.200$ | 36.48±1.66 |
| *Left* | *0.095±0.007* | *0.082* | *0.112* | *27.46%* | | |
| *Right* | *0.095±0.007* | *0.080* | *0.116* | *31.29%* | | |
| **ME** | 0.146±0.011 | 0.122 | 0.174 | 29.95% | $F(8,101) = 2.819$, $p = 0.288$ | 27.90±1.33 |
| *Left* | *0.073±0.006* | *0.062* | *0.085* | *27.52%* | | |
| *Right* | *0.073±0.005* | *0.060* | *0.089* | *32.29%* | | |
| **LO** | 0.045±0.004 | 0.038 | 0.054 | 29.94% | **$F(8,101) = 2.733$, $p = 0.009$** | 8.58±0.43 |
| *Left* | *0.0223±0.002* | *0.0179* | *0.027* | *33.96%* | | |
| *Right* | *0.0224±0.002* | *0.0185* | *0.027* | *30.72%* | | |
| **CX** | 0.0033±0.0004 | 0.0021 | 0.0043 | 51.74% | $F(8,101) = 1.839$, $p = 0.400$ | 0.63±0.07 |
| **OTH** | 0.166±0.019 | 0.100 | 0.203 | 50.61% | $F(8,101) = 1.643$, $p = 0.433$ | 31.78±2.06 |

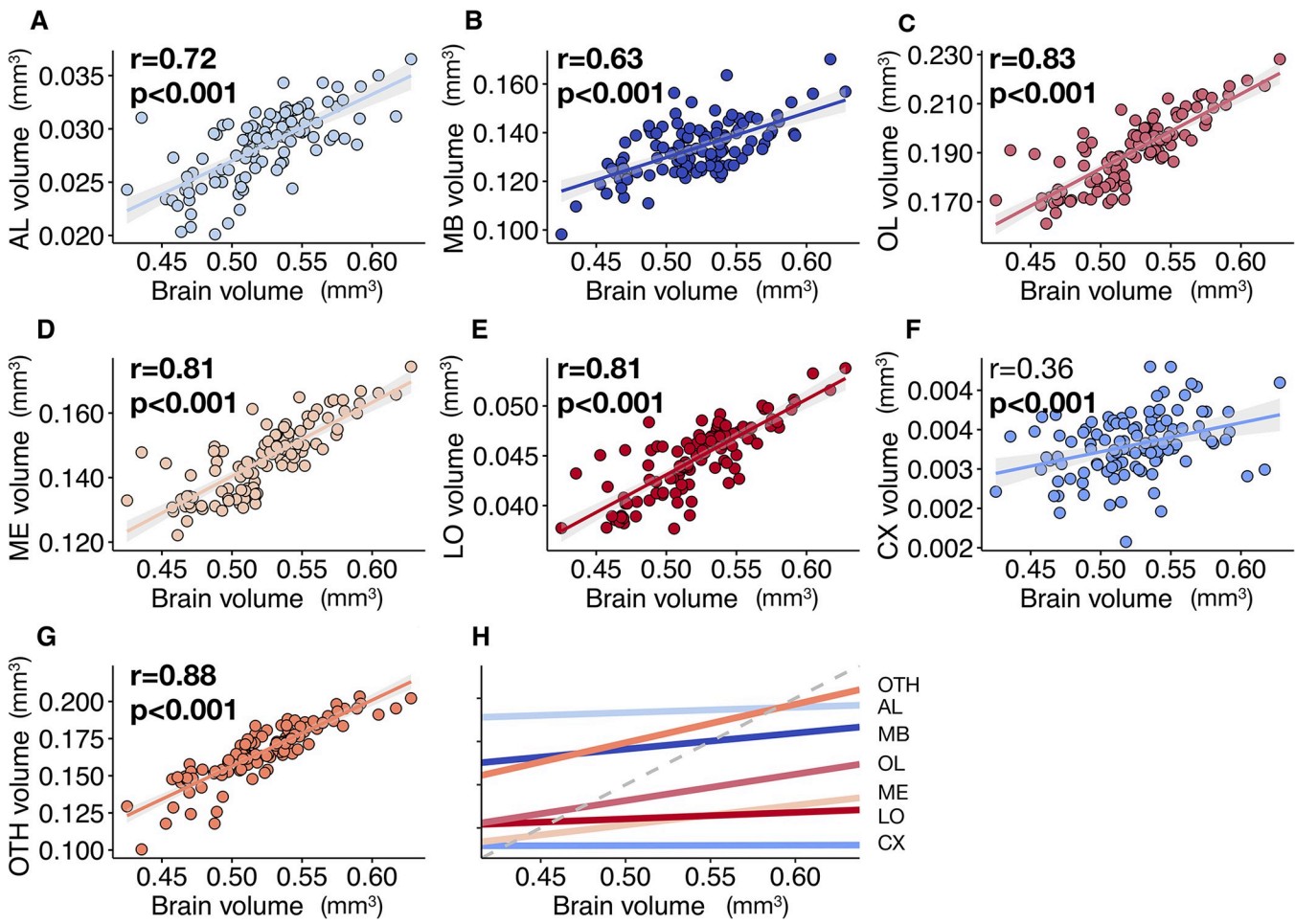

**Fig 5. Correlation between neuropil volumes and total brain volume (mm³) for honey bees (_N_ = 110).** (**A**) Antennal lobes (AL). (**B**) Mushroom bodies (MB). (**C**) Optic lobes (OL) (=ME+LO). (**D**) Medullae (ME). (**E**) Lobulae (LO). (**F**) Central complex (CX). (**G**) Other neuropils (OTH). Regression lines displayed with 95% confidence intervals. Pearson correlation coefficient ($r$) and p-value are given. Strong correlations ($r$>0.40) and significant correlations ($p$<0.05) are displayed in bold. (**H**) Linear correlations for the different neuropils (y-axis not given: differs for each neuropil). The grey dashed line indicates true isometric correlation (slope = 1).

dataset, however, AL volume is positively correlated with all other neuropils. We also found positive correlations for MB, CX and OTH with all other neuropils, except between MB and CX, which were not correlated. We searched for relevant allometric relationships (S3 Table). Again, we found strong correlations between the relative volumes of OL, LO and ME, consistent with previous studies [29]. There was a negative correlation between the relative volumes of AL and MB, while a previous study [29] reported a positive correlation between the relative volumes of AL and MB lobes (but not the calyces), but in our study we did not distinguish the structures within MBs. In addition, the relative volume of AL was positively correlated with those of OL, ME, and LO, while previous work on honey bees [29] and other bee species [44] suggested a negative correlation between AL and OL as a possible trade-off between visual and olfactory processing. Also, the relative OTH volume was negatively correlated with all other neuropils except AL. Overall, honey bees with larger brains tend to have larger neuropils, with the exception of CX, which is not as strongly related to brain size as the other neuropils. This weaker correlation could also be due to the small size of the neuropil and potential measure uncertainties caused by the difficulty of its segmentation.

## Variance in honey bee brain volumes was similar within and between colonies

Next, we explored the distribution of honey bee brain volumes within and between colonies. Social insect colonies can be considered superorganisms, characterised by a division of labour among workers that is partly determined by the genetics, age and morphology of individuals [45]. In honey bees, MB volume increases with age [24] and foraging experience [40,46], and AL volume correlates with behavioural tasks, i.e. nurse, comb-builder, nectar or pollen forager [47]. These variations are thought to support division of labour, as larger volumes (relative and/or absolute) of specific neuropils may allow some individuals to be more efficient at certain tasks or result from behavioural specialisations. For example, larger ALs and/or MBs may provide foragers with better abilities at learning spatial information and floral cues than in-hive workers [40]. While some studies investigated variations in brain size between species [28] or within species due to stress exposure (e.g. pesticides [13], nutrition [37]), to our knowledge, the magnitude of intra- and inter-colony variability under standard conditions has not yet been investigated. Stable patterns of within-hive variation in brain volumes across colonies with different genetic backgrounds and in different geographic areas could suggest selection for variability.

In our study, honey bees in the nine colonies exhibited overall similar average brain volumes (F-test: $p = 0.428$; Fig 6A and Table 1), hence, inter-colony differences do not explain the substantial (32%) inter-individual variability. Within colonies, the values were also relatively stable (S4 Table). However, several neuropils exhibited significant variations in their absolute volumes across colonies (Fig 6 and Table 1): AL (F-test: $p<0.001$; Fig 6B), MB (F-test: $p = 0.025$; Fig 6C), and LO (F-test: $p = 0.009$; Fig 6F). Interestingly, intra-colony variability of neuropils was generally lower and CX and OTH were the more variable (resp. 11 to 49% and 11 to 51%, depending on the colony), despite non-significant changes between colonies (S4 Table and Fig 6G and 6H). Notably, we only found low intra-colonial variations in MB volumes (7 to 30%), which are known to increase in size by 15% with age and foraging experience over the lifespan [24]. This low level of variability is probably due to the fact that we only studied honey bee foragers (see "Methods") and therefore could not reflect the reported brain plasticity between emerging workers, nurses and foragers. Thus, overall, we found moderate variability in brain size and neuropil volumes that were rather stable within colonies, based on a relatively homogeneous sample regarding strain (Buckfast) and behavioural specialisation (foragers).

## Bumblebees showed higher variability in brain and neuropil volumes than honey bees

To demonstrate the generalisability of our approach to other sample series and illustrate how this can be used to compare species, we applied the same analyses to the bumblebee brains. Bumblebees are increasingly used for comparative behavioural and cognitive research [48] and a brain atlas was recently published [49]. In contrast to honey bees, division of labour in bumblebees is primarily based on body size variation [50] with little effect of age (foragers tend to be larger than non-foragers [51]). Therefore, it is not surprising that overall bumblebees exhibited higher variation levels in total brain volume (52%; Fig 7A and S5 Table) and neuropil volumes (44 to 85%; Fig 7B–7H), as compared to honey bees. Note, however, that the 77 bumblebees randomly sampled in the four source colonies were likely to be more heterogeneous in terms of behaviour than the 110 honey bees, which were all foragers. Yet, all neuropils scaled isometrically with total brain volume (S9 Fig), with again only a lower correlation for CX ($r = 0.59$, $p<0.001$). Similar to honey bees, we found that only the relative volume (S5 Table) of

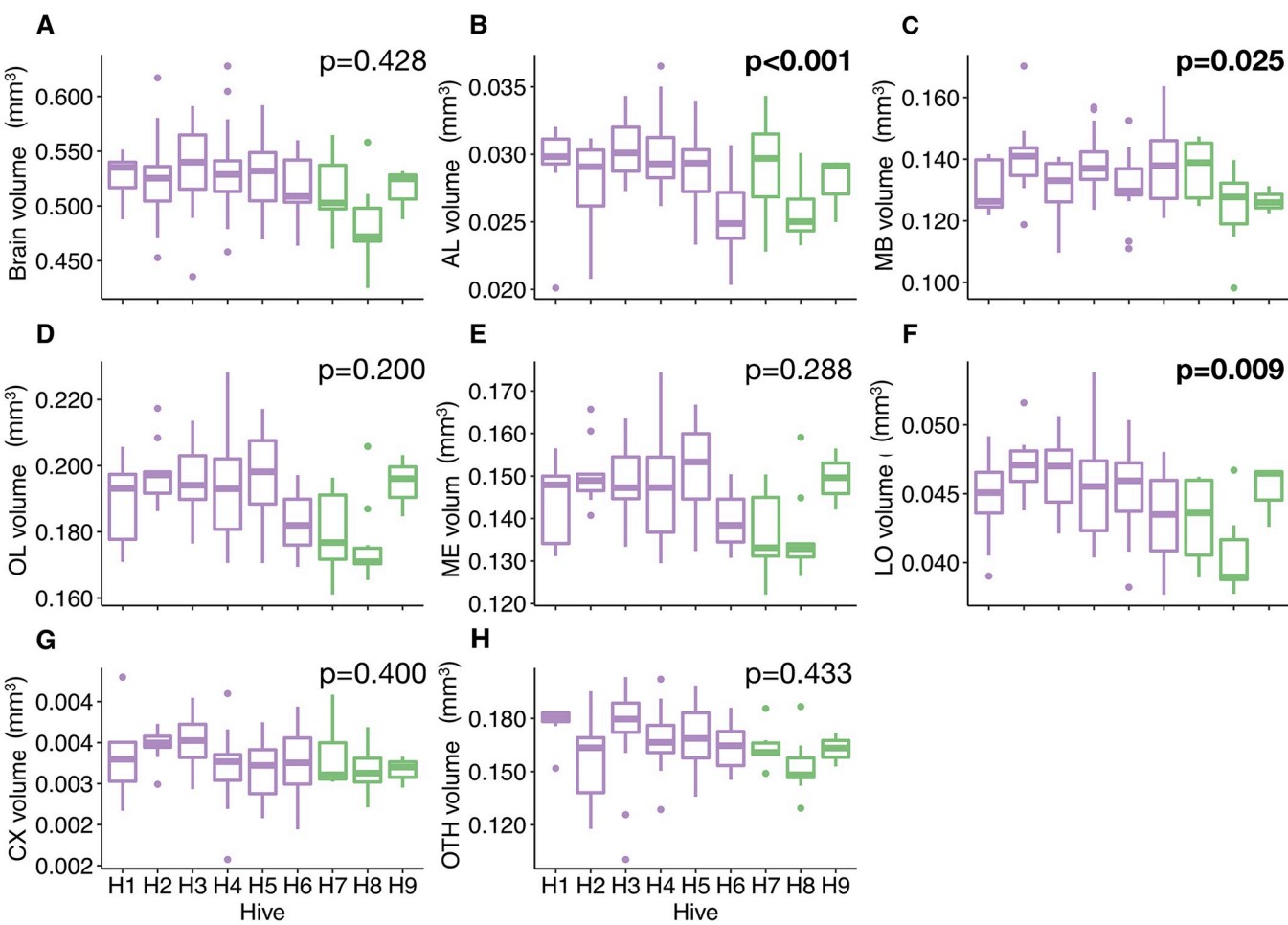

**Fig 6. Variation in brain volume and absolute volume of neuropils (mm³) between honey bee colonies (population A - *purple*, N = 6 hives; population B - *green*, N = 3 hives).** (**A**) Total brain. (**B**) Antennal lobes (AL). (**C**) Mushroom bodies (MB). (**D**) Optic lobes (OL). (**E**) Medullae (ME). (**F**) Lobulae (LO). (**G**) Central complex (CX). (**H**) Other neuropils (OTH). Boxplots show median volumes (intermediate line) and quartiles (upper and lower lines). Statistical comparisons (p-values) for the neuropil volumes between hives were obtained from the F-test following LMMs and are displayed in bold when significant (see sample sizes in the Methods and statistical details in Table 1).

OTH was positively correlated with total brain volume and the relative volumes of LO, ME (and consequently OL) were negatively correlated with total brain volume (S10 Fig).

Regarding correlations between neuropil volumes (S6 Table), we found significant and strong positive effects for all possible permutations in absolute volumes. Relative volumes showed the same major trends as in honey bees (S6 Table): OL, ME and LO were positively correlated with each other, OTH was negatively correlated with all other neuropils, except CX, and relative AL volume was positively correlated with MB, OL, ME and LO relative volumes. Thus, the larger the brain, the larger the proportion of OTH and the relatively smaller the other neuropils.

The four bumblebee colonies did not show significant differences for any of the neuropil volumes, except for LO (S5 Table), while we found volume differences for AL, MB, OL and LO when comparing honey bee colonies (Table 1 and Fig 6). Again, this is not surprising since the bumblebee sample was more heterogeneous in terms of individual size than the sample of forager honey bees. Thus, like for honey bees, we found relatively stable intra-colony variability in brain and neuropil volumes (S5 Table).

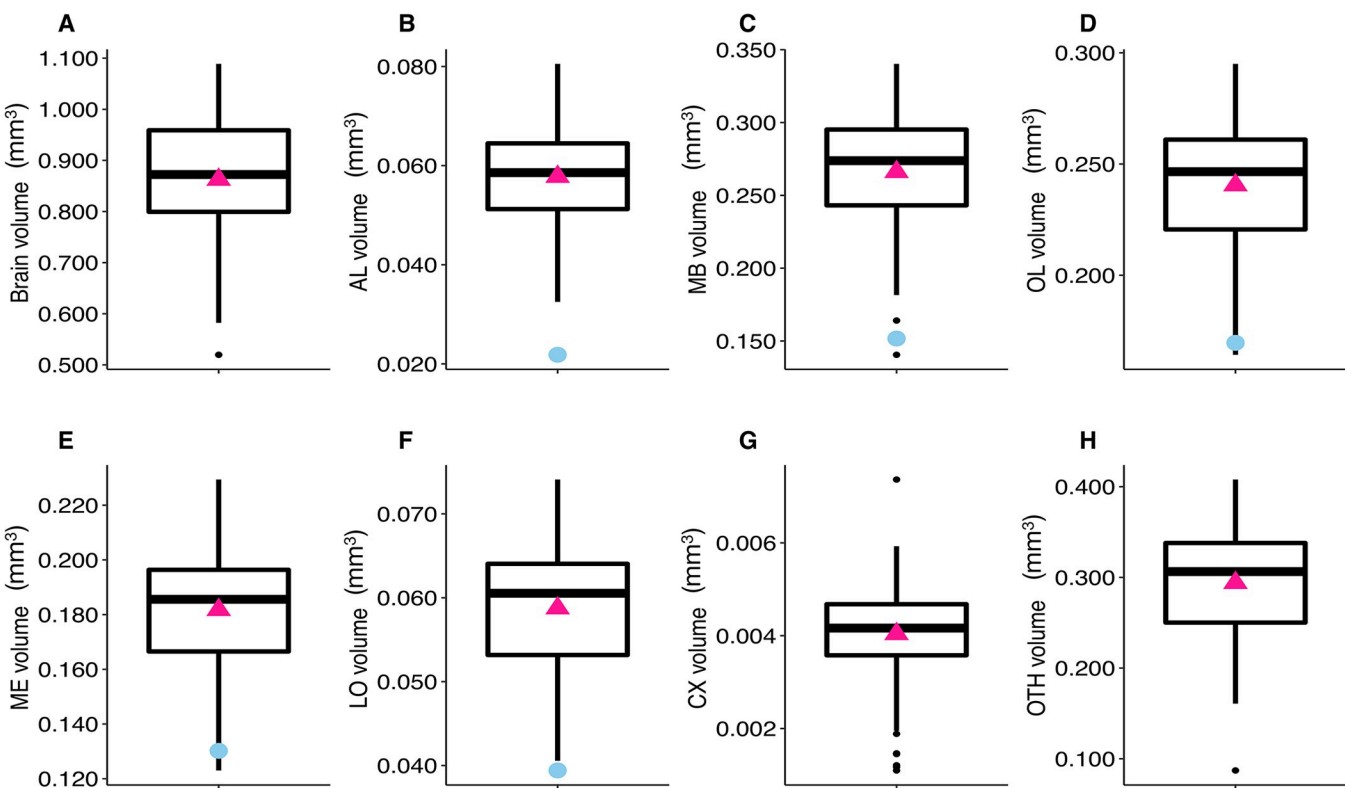

**Fig 7. Variation in brain volume and neuropils volumes (mm³) and correlation between neuropil volumes and total brain volume for bumblebees** (*N* = 77). (**A**) Total brain. (**B**) Antennal lobes (AL). (**C**) Mushroom bodies (MB). (**D**) Optic lobes (OL). (**E**) Medullae (ME). (**F**) Lobulae (LO). (**G**) Central complex (CX). (**H**) Other neuropils (OTH). Boxplots show median volumes (intermediate line) and quartiles (upper and lower lines). Pink triangles display mean volumes. Coloured symbols show the mean of neuropil volumes described for bumblebees in another study using CT scan: Smith et al. [14] (*N* = 38 bees - *blue*).

## Optic lobes showed left-right asymmetry in bumblebees but not in honey bees

Bees have lateralized cognitive abilities, consistently learning better with their right eye [52] and right antenna [53–55]. However, no study has yet reported anatomical asymmetries in the brain that might support these observations [56,57]. Taking advantage of our large number of complete bee brain volumes, we compared the left and right volumes of all paired neuropils in the two bee species (Tables 1 and S5 and Fig 8). The proximity of left and right sides in MB structures posed difficulties in their post-segmentation separation, resulting in a comparison based on only 59 honey bees and 36 bumblebees. However, there is potential for improvement in the future by considering a separation of the left and right MB structures in the training data itself.

First, we tested for lateralization of individual brain areas. For honey bee brains, we found no lateral differences in the absolute volumes of AL (LMM: *p* = 0.324), MB (LMM: *p* = 0.243), ME (LMM: *p* = 0.964), LO (LMM: *p* = 0.285) and OL (LMM: *p* = 0.736). We did not find any significant difference in the relative volume of AL (LMM: *p* = 0.341), MB (LMM: *p* = 0.338), ME (LMM: *p* = 0.907), LO (LMM: *p* = 0.245) and OL (LMM: *p* = 0.665) neither. By contrast, the same analysis revealed some asymmetry in bumblebee brains. While there was no difference in the lateral absolute volumes of AL (LMM: *p* = 0.190) and MB (LMM: *p* = 0.664), the left ME (LMM: *p* = 0.019) and left LO were significantly smaller (LMM: *p* = 0.002) in 62% and

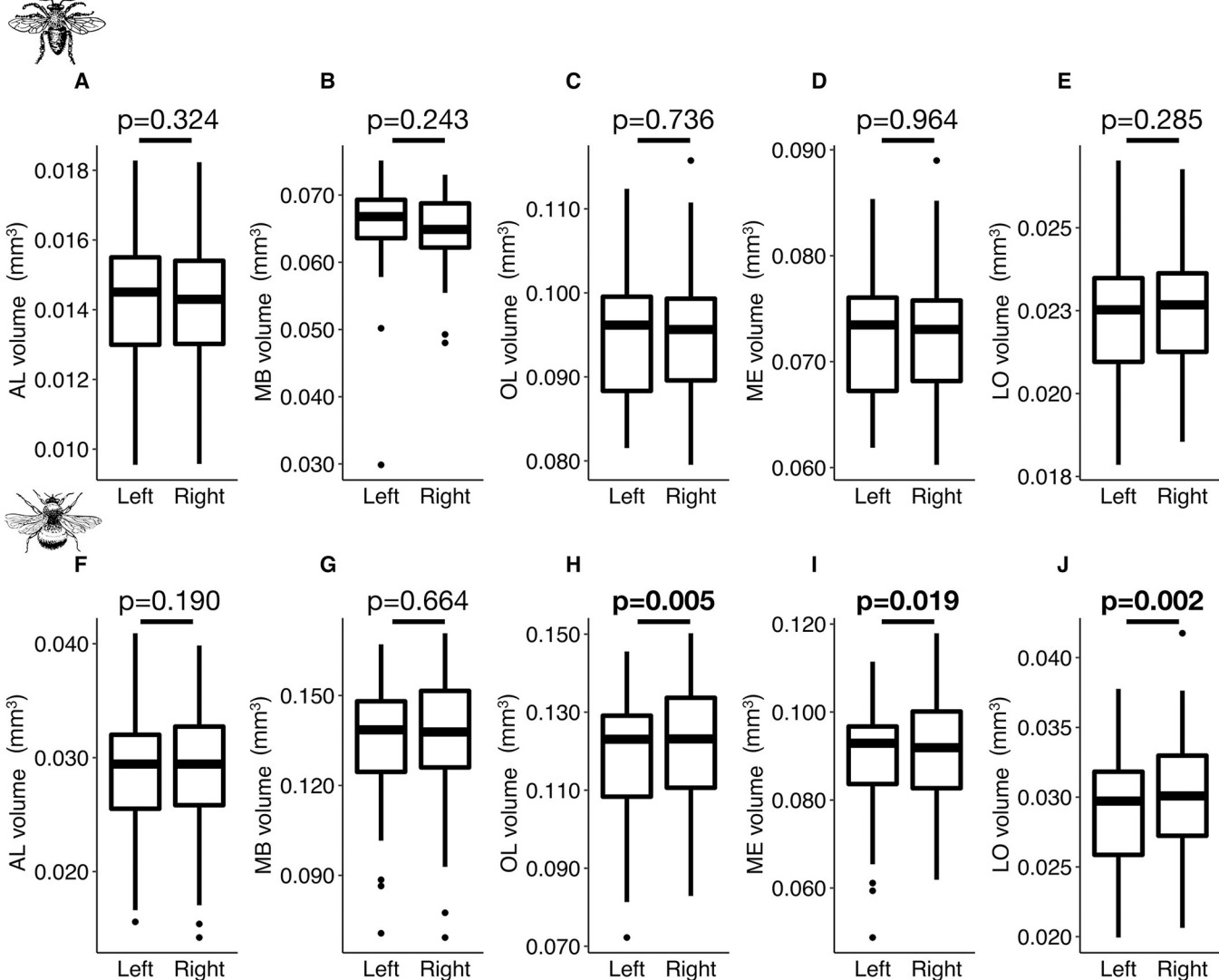

**Fig 8. Comparisons between left and right volumes (mm³) for paired neuropils in honey bees (A−E) and bumblebees (F−J).** (A and F) Antennal lobes (AL) (*N* = 110 honey bees, *N* = 77 bumblebees). (**B** and **G**) Mushroom bodies (MB) (*N* = 59 honey bees, *N* = 36 bumblebees). (**C** and **H**) Optic lobes (OL) (*N* = 110 honey bees, *N* = 77 bumblebees). (**D** and **I**) Medullae (ME) (*N* = 110 honey bees, *N* = 77 bumblebees). (**E** and **J**) Lobulae (LO) (*N* = 110 honey bees, *N* = 77 bumblebees). Boxplots show median volumes (intermediate line) and quartiles (upper and lower lines). Statistical differences (p-values) for the neuropil volume between left and right side were obtained with Student's paired samples t-Test, and are displayed in bold when significant.

66% of the brains, respectively, resulting in an overall smaller left OL (LMM: *p* = 0.005) in 62% of the brains (Fig 8). We found similar results when analysing the relative lateral volumes with no differences in AL (LMM: *p* = 0.247) and MB (LMM: *p* = 0.864), but smaller left ME (LMM: *p* = 0.015) and left LO (LMM: *p* = 0.001), which also results in the left OL having a smaller relative volume than the right side (LMM: *p* = 0.004).

We next investigated potential whole-brain lateralization by comparing the asymmetries along the right-left axis between all paired neuropils (Fig 9). Firstly, we looked for correlations between AL and OL (Fig 9A and 9D). In honey bees, we found an even distribution of individuals across the four potential brain classes: larger right AL and right OL (24.5%), larger right AL and smaller right OL (22.7%), smaller right AL and larger right OL (24.5%), smaller right AL and OL (28.2%). In contrast, we found more bumblebees with larger right AL and larger

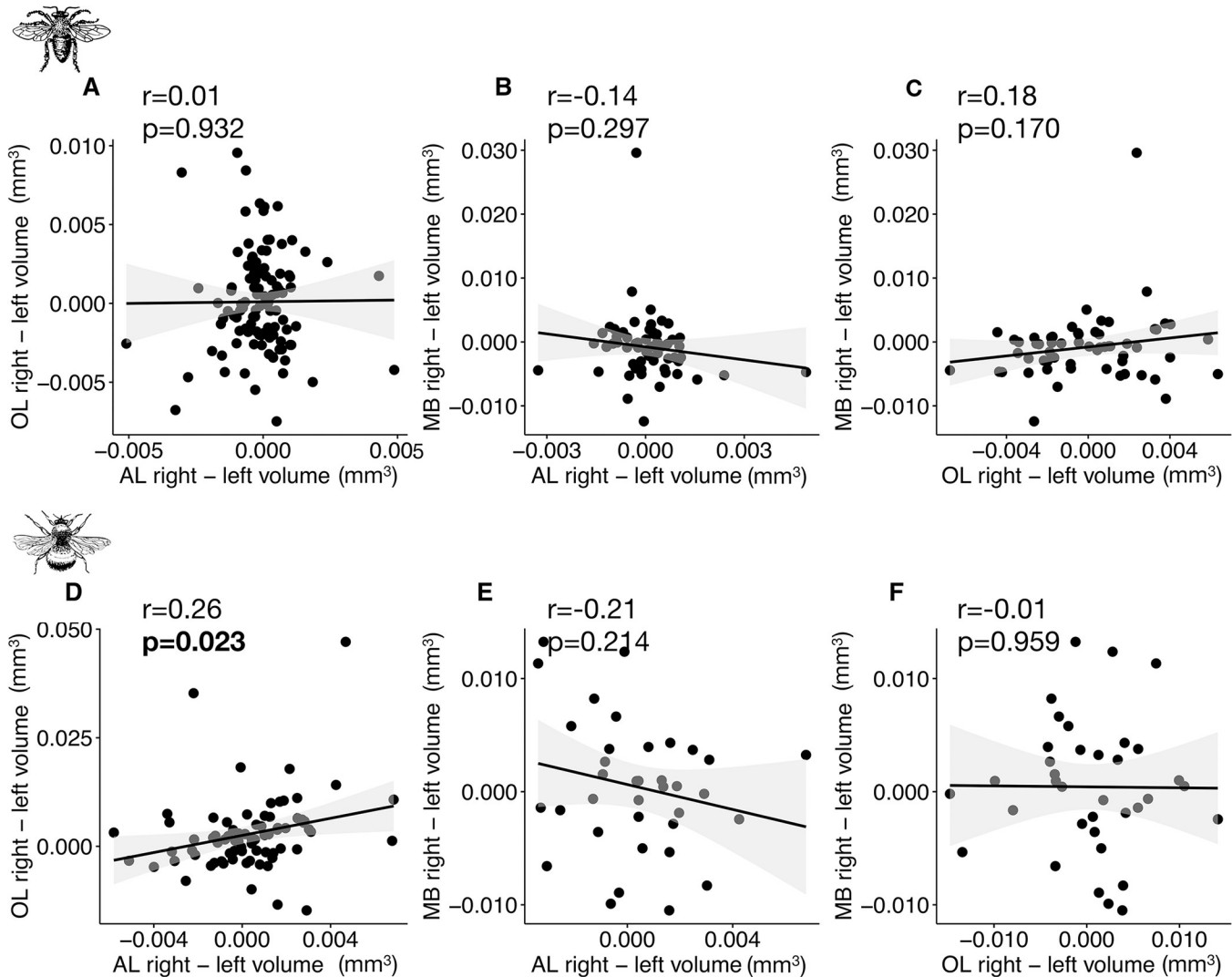

**Fig 9. Correlations between right-left volumes (mm³) between AL, OL and MB.** (**A** and **D**) Correlations between AL right-left and OL right-left volumes (*N* = 110 honey bees, *N* = 77 bumblebees). (**B** and **E**) Correlations between MB right-left and AL right-left volumes (*N* = 59 honey bees, *N* = 36 bumblebees). (**C** and **F**) Correlations between MB right-left and OL right-left volumes (*N* = 59 honey bees, *N* = 36 bumblebees). Pearson correlation coefficient (r) and p-value are given. Strong correlations (*r*>0.40) and significant correlations (*p*<0.05) are displayed in bold.

right OL (39.0%) than in the other categories (S11 Fig). Additionally, we found a positive correlation between the right-left AL and OL volumes (*r* = 0.26, *p* = 0.023; Fig 9D). When taking into account the MB in the analyses, we found no significant correlations between MB and AL or MB and OL in honey bees (Fig 9B and 9C) or bumblebees (Fig 9E and 9F).

Therefore, there was no clear lateralization at the brain level, except in bumblebees where most individuals had larger right AL and OL. This anatomical asymmetry could contribute to the well-documented inter-individual cognitive variability in bumblebees [58] and the fact that bees have better visual and olfactory learning abilities when stimuli are applied to right sensory organs [52–55]. However, whether this correspondence between neuroanatomy and behaviour is causal remains to be explored [59]. The fact that we did not find such evidence of volumetric asymmetries in honey bees, despite well-known learning asymmetries [52,53], may indicate that left-right differences in ALs or OLs were too small to reach significance in our sample.

Alternatively, the reported lateralized learning may be caused by differences between right and left sensory organs (e.g. there are more olfactory sensillae on the right antenna [44]), instead of brain structures.

## Conclusion

Comparative analyses of animal brain volumes are currently limited to a few individuals in model species [56]. Here we show how deep neural networks can be used for fast and accurate analysis of large amounts of volumetric brain images, opening new possibilities for broad comparative analyses in insects and invertebrates in general. Additionally, this method can complement other histological procedures commonly used to study neuropiles, such as immunostaining and Lucifer yellow, by potentially reducing the number of required training images through improved contrast. Future investigations could explore and publish on the comparative performance of various neuropil markers and imaging techniques, including confocal microscopy [60]. Our results on two social bee species show high natural variability in brain area sizes that is stable across colonies. These anatomical differences could potentially generate behavioural variability and support adaptive collective behaviour, thereby calling for future studies in more species with contrasted social ecologies. Brain asymmetries may also explain previously reported behavioural lateralization [57,61]. Future combination of brain and behavioural analyses using our method for automated 3D image segmentation will help address important open and hotly debated questions related to neuroscience and cognition. Are bigger brains more performant [62]? What are the influences of social and ecological factors in brain evolution [2]? What is the effect of environmental stressors on brain development and cognition [13]? Our 3D image analysis tool, Biomedisa, is accessible via a web browser and does not require complex or tedious software configuration or parameter optimisation, making it user-friendly even for biologists with no substantial computational expertise. We therefore expect our approach to be widely used to study the evolution of animal body parts or organs, beyond the brain itself. Indeed, recent studies showed Biomedisa is suitable for many types of volumetric image data other than micro-CT scans, e.g. from confocal laser scanning microscopy (CLSM), focused ion beam scanning electron microscopy (FIB-SEM), histological imaging [63], or magnetic resonance imaging (MRI) [64], which offer the opportunity for a wide range of studies in biology, ecology and evolution to perform large-scale, cost-effective and time-efficient analyses.

## Materials and methods

### Sample preparation and CT scan

We performed micro-computed tomography scanning of honey bees (*A. mellifera*, Buckfast) and bumblebees (*B. terrestris*). Honey bees were collected from 9 hives located in two apiaries around Toulouse, France, in August 2020 (Population A (GPS coordinates: 43.55670, 1.47073): 100 bees from 6 hives: H1 ($N$ = 9), H2 ($N$ = 11), H3 ($N$ = 18), H4 ($N$ = 19), H5 ($N$ = 17), H6 ($N$ = 16); population B (43.26756, 2.28376): 20 bees from 3 hives: H7 ($N$ = 6), H8 ($N$ = 11), H9 ($N$ = 3)). Foragers returning to the colony were collected at the hive entrance, frozen and stored at -18˚C. Bumblebees were purchased from a commercial supplier (Koppert, France) and 77 workers were collected from 4 hives and frozen at -18˚C: H1 ($N$ = 18), H2 ($N$ = 20), H3 ($N$ = 21), H4 ($N$ = 18)). The analysed honey bees and bumblebees were of unknown age, and no body size information was available for these specimens. While all honey bees were strictly foragers, bumblebees were randomly collected and may therefore constitute a more heterogeneous population. For staining bee brains, we removed the front cuticle just above the mandibles and submerged the heads in phosphotungstic acid (5% in a 70/30%

ethanol/water solution) for 15 days at ambient temperature [14]. The staining agent is non-hazardous and does not lead to overstaining of the soft tissues, in contrast to other compounds previously used in CT scan studies of insect brains such as osmium-tetroxide [42] or iodine [65]. Two heads were scanned at the same time (as both fit in the field of view of the flat-panel imager) using the micro-CT station EasyTom 150/RX Solutions (Montpellier Ressources Imagerie, France), with the following parameters: resolution of 5.4 μm isotropic voxel size, 40 kV, 130 μA, 736 radiographic projections (acquisition time: 15 minutes). Raw data for each brain scan was reconstructed using X-Act software (RX Solutions, Chavanod, France). The reconstructed volumes were then re-oriented to the same (frontal) plane-of-view and each brain was re-sliced into a new series of two-dimensional images.

## Statistical analysis

We analysed the parameters obtained from the reconstructed neuropils using R Studio v.1.2.5033 [66]. Brain volumes were normally distributed. We assessed correlations between brain neuropil volumes using the *rcorr* function from the *Hmisc* package [67]. To analyse the inter-colonial variations of brain volume, we conducted linear mixed models (LMMs) using the *lme4* package [68], with hive as fixed effect and population as random factor. LMMs were followed by F-tests to test the significance of fixed categorical variables using the *anova* function in the *car* package [69]. To assess the potential lateralization of paired neuropils, we separated left and right sides in an automatised post-processing step and conducted Student's paired samples t-Test using Python's SciPy library. The script is included in the Biomedisa GitHub repository.

## Manual processing times with or without Biomedisa

A conventional manual segmentation of a CT scan took about 5 to 6 h, and 2 to 3 h with Biomedisa's semi-automated segmentation. Within the automatic segmentation process, the manual effort averaged 5 to 10 min. That means, the manual segmentation of 110 honey bee scans took 550 to 660 h, while semi-automatic creation of three initial 3D training images and the manual processing of the automated segmentation results of the remaining 107 CT scans took at least 15 h (3×2×60 min+107×5 min) and a maximum of 27 h (3×3×60 min+107×10 min). For the segmentation of 77 CT scans of bumblebee brains, manual segmentation took 385 to 462 h. Here, the creation of 13 three-dimensional training images and the manual processing of the remaining 64 automated segmentation results took at least 31 h (13×2×60 min+64×5 min) and up to 50 h (13×3×60 min+64×10 min).

## Network architecture and parameters

Biomedisa uses Keras with TensorFlow backend. A patch-based approach is utilised in which 3D patches of the volumetric images are used instead of the entire CT scan. The patches serve as input for a 3D U-Net and have a size of 64×64×64 voxels. Before extracting the patches, 3D images are scaled to a size of 256×256×256 voxels and transformed to have the same mean and variance. An overlapping of the patches is achieved by a stride size of 32 pixels that can be configured in Biomedisa. The network architecture of the deep neural network follows the typical architecture of a 3D U-Net [70]. It consists of a contracting and an expansive part with a repeated application of two 3×3×3 convolutions, each followed by batch normalisation and a rectified linear unit (ReLU) activation layer. Each contracting block is followed by a 2×2×2 max pooling operation with stride size of 2 for downsampling. At each downsampling step, the number of feature channels is doubled, commencing with 32 channels and culminating at 1024. Every step in the expansive part consists of an upsampling of the feature map and a

concatenation with the corresponding cropped feature map from the contracting path, followed by two 3×3×3 convolutions, with each followed by batch normalisation and a ReLU activation layer. At the final layer, a 1×1×1 convolution is used to map each feature vector to the desired number of labels. To train the network, stochastic gradient descent is used with a learning rate of 0.01, decay of $1\times10^{-6}$, momentum of 0.9, enabled Nesterov momentum, 200 training epochs, and a batch size of 24 samples.

### Removing outliers

To evaluate the segmentation results, Biomedisa's cleanup function was used for all neuropils except CX to automatically remove outliers or islands with a threshold of 0.1. This threshold value was set lower than the standard configuration (0.9) in order to avoid a partial deletion of paired neuropils.

### Evaluating the creation of training data

To evaluate the semi-automatic segmentation used to create the labels for the 3D training images, we compared the commonly used linear interpolation in AVIZO 2019.1 and the smart interpolation in Biomedisa using the set of 26 three-dimensional training images. For intervals including CX we used every 5th slice of the ground truth data, otherwise every 10th slice as initialisation. Using the same pre-segmented slices, Biomedisa's smart interpolation, which also takes into account the underlying 3D image data, achieves higher segmentation accuracy (average Dice score of 0.967) compared to the conventional morphological interpolation of AVIZO 2019.1 (average Dice score of 0.928, Fig 3 and S1 Table). Thus, the manual work required to create the training data for a neural network is significantly reduced.

### Increasing the number of 3D training images

To test the performance of the automatic segmentation in terms of the number of 3D training images, we evaluated the accuracy of the trained network using 3 to 26 three-dimensional training images (Fig 3 and S1 Table). Here, the 110 three-dimensional honey bee micro-CT scans and the corresponding labels were split into 26 three-dimensional training images, 30 three-dimensional validation images and 54 three-dimensional test images. While the conventional practice often involves allocating approximately 80% of the data for training and 20% for testing or using a split of 60% for training and 20% each for validation and testing, we opted for a training dataset size that aligns with the analysis procedure for our specific bee brain dataset. Thus, we used up to 26 CT scans for training, selecting a similar amount for validation, considering the compute-intensive nature of validation during training, and assigned the remaining 3D images for testing. In instances where honey bees were initially scanned upside down, the 3D image data was flipped along the z-axis. The accuracy of the training process, measured using the Dice score, was evaluated using the validation data after each of the 200 epochs performed. Only the best performing network was saved. Finally, the trained network was applied to the 54 three-dimensional test images and the average Dice score of the segmentation results was calculated. While increasing the number of 3D training images improves the accuracy of the automatic segmentation and thus reduces the required manual post-processing, this improvement gradually slows down with an increasing number of 3D training images. Therefore, one must weigh the gain in accuracy against the additional effort required. Using the honey bee dataset, 12 to 20 three-dimensional training images are sufficient for adequate automatic segmentation. Comparable results are achieved by utilising 3 to 26 three-dimensional training images from the bumblebee dataset, accompanied by 20 three-dimensional validation, and 24 three-dimensional test images (S3 Fig).

## Cropping 3D image data to the region of interest improves accuracy

The network was also trained and evaluated using uncropped 3D images (i.e. original honey bee CT scans). Cropping the 3D image data to the area of the neuropils significantly increased the segmentation accuracy of the neural network from a total Dice score of 0.928 for the uncropped 3D images to 0.970 for the cropped 3D image data (Fig 3 and S1 Table). By default, Biomedisa scales each 3D image to a size of 256 pixels for each axis to facilitate training. The cropped 3D image data (average size of 451×273×167 voxels) were thus scaled with an averaged factor of 0.57 and 1.53 along the x- and z-axes, and only marginally along the y-axis. Without cropping the 3D image data (average size of 844×726×485 voxels), a large amount of redundant information is added to the training data. In addition, the loss of resolution due to the scaling of the 3D image is significantly larger compared to cropped 3D image data.

## Automatic cropping

As an alternative to manual cropping, a neural network can be trained with Biomedisa to automatically crop the 3D image data to the region of interest before segmentation. Here the DenseNet121 [71] pre-trained on the ImageNet database is used to decide whether a slice of the volume contains the object or not. Applying the network to all three axes creates a bounding box covering the honey bee brain. Adding a buffer of 25 slices to all sides after cropping ensures that the entire object is within the bounding box. The network is trained in two steps. First, the head of the pre-trained network is removed and replaced with a global average pooling layer, a dropout layer with a probability of 0.3 to avoid overfitting, and a final dense layer with sigmoid activation function. The weights of the base model are kept fixed while the head is optimised with binary cross-entropy, Adam optimiser with a learning rate of 0.001 and 50 epochs. Second, the entire DenseNet121 is fine-tuned to the honey bee training data with a learning rate of $1 \times 10^{-5}$ and 50 epochs. Auto-cropping achieves a Dice score of 0.952, which increases accuracy by 2.4% compared to uncropped honey bee 3D image data (Fig 3 and S1 Table).

## Evaluation metrics

For two segmentations $X$ and $X'$ consisting of $n$ labels, the Dice similarity coefficient (Dice) is defined as

$$\text{Dice} = \frac{2 \sum_{i=1}^{n} |X_i \cap X_i'|}{|X| + |X'|},$$

where $|X|$ and $|X'|$ are the total number of voxels of each segmentation, respectively, and $X_i$ is the subset of voxels of $X$ with label $i$. For the surfaces $S_i$ and $S_i'$ of the two segmentations, the average symmetric surface distance (ASSD) is defined as

$$\text{ASSD} = \frac{1}{|S| + |S'|} \sum_{i=1}^{n} \left( \sum_{p \in S_i} d(p, S_i') + \sum_{p' \in S_i'} d(p', S_i) \right),$$

Where

$$d(p, S_i') = \min_{p' \in S_i'} \| p - p' \|_2$$

is the Euclidean distance from a point $p$ on the surface $S_i$ of label $i$ to the closest point $p'$ on the corresponding surface $S_i'$ of the second segmentation.

### Denoising 3D image data

Volumetric images are denoised using an arithmetic mean filter

$$A(x, y, z) = \frac{1}{|M|} \left( \sum_{i,j,k \in M} I(i, j, k) \right)$$

with a filter mask $M$ of 3×3×3 voxels.

## Supporting information

**S1 Appendix. Metadata of collected honey bees and bumblebees.** This table contains comprehensive data, including population, collection date, hive information, and specimen ID. Moreover, it presents absolute and relative volumes of various brain areas (MB, CX, AL, ME, LO, OL, and OTH), along with details on total brain volume and the sizes of the left and right sides of paired neuropils.
(XLSX)

**S1 Fig. Segmentation results (left) and manually corrected results with segmentation errors highlighted (right) of Biomedisa's deep neural network trained on 26 honey bee CT scans.** (**A**) Correct segmentation without errors (bee ID 87, hive H4). (**B**) Partly flawed segmentation results with a typical outlier on the right edge of the image (bee ID 64, hive H5, segmentation accuracy: ME 97.6%, total 99.1%). (**C**) Significantly flawed segmentation result (bee ID 98, hive H6, total segmentation accuracy 87.7%).
(DOCX)

**S2 Fig. Segmentation results (left) and manually corrected results with segmentation errors highlighted (right) of Biomedisa's deep neural network trained on 13 bumblebee CT scans.** (**A**) Correct segmentation without errors (ID 93). (**B**) Partly flawed segmentation results, with CX almost not recognised (ID 81, segmentation accuracy: CX 20.4%, total 97.9%). (**C**) Significantly flawed segmentation result (ID 57, total segmentation accuracy 88.3%).
(DOCX)

**S3 Fig. Fine-tuning neural network on bumblebee data for an increasing number of 3D training images.** Results from 3 to 26 three-dimensional bumblebee training images, utilising 20 three-dimensional validation images during training. The final evaluation is performed on 24 three-dimensional test images. The networks pre-trained on honey bee image data and fine-tuned to bumblebee image data (*green*), showing improved performance for up to 7 three-dimensional training images compared to training from scratch with initially standard normal distributed weights (*red*).
(DOCX)

**S4 Fig. Bumblebee training and validation data accuracy.** Results from 13 three-dimensional training images and 20 three-dimensional validation images show the progress of accuracy in bumblebee data. While the standard accuracy of the training data (*red*) continues to improve over the course of training, the standard accuracy (*green*) and the Dice score (*blue*) of the validation data level off at their maximums of 0.975 after 154 epochs (*orange*) and 0.953 after 129 epochs (*cyan*), respectively. Dice score for training data is not available on Biomedisa.
(DOCX)

**S5 Fig. Bumblebee training and validation loss.** Results from 13 three-dimensional training images and 20 three-dimensional validation images. While the loss of the training data (*red*) continues to decrease over the course of training, the loss of the validation data (*green*) begins

to increase marginally after reaching its minimum at 0.08 (*orange*).
(DOCX)

**S6 Fig. Honey bee training and validation data accuracy.** Results from 26 three-dimensional training images and 30 three-dimensional validation images show the progress of accuracy in honey bee data. While the standard accuracy of the training data (*red*) continues to improve over the course of training, the standard accuracy (*green*) and the Dice score (*blue*) of the validation data level off at their maximums of 0.985 after 197 epochs (*orange*) and 0.969 after 123 epochs (*cyan*), respectively. Please note that the Dice score for training data is not available on Biomedisa.
(DOCX)

**S7 Fig. Honey bee training and validation loss.** Results from 26 three-dimensional training images and 30 three-dimensional validation images. While the loss of the training data (*red*) continues to decrease over the course of training, the loss of the validation data (*green*) begins to increase marginally after reaching its minimum at 0.05 (*orange*).
(DOCX)

**S8 Fig. Correlation between relative volumes of neuropils and total brain volume (mm$^3$) for honey bees (N = 110).** (**A**) Antennal lobes (AL). (**B**) Mushroom bodies (MB). (**C**) Optic lobes (OL). (**D**) Medullae (ME). (**E**) Lobulae (LO). (**F**) Central complex (CX). (**G**) Other neuropils (OTH). Regression lines displayed with 95% confidence intervals. Pearson correlation coefficient (r) and p-value are given. Strong correlations ($r>0.40$) and significant correlations ($p<0.05$) are displayed in bold. (**H**) Linear correlations for the different neuropils relative volume (y-axis not given: differs for each neuropil). The grey dashed line indicates true isometric correlation (slope = 1).
(DOCX)

**S9 Fig. Correlation between neuropil volumes and total brain volume (mm$^3$) for bumblebees (N = 77).** (**A**) Antennal lobes (AL). (**B**) Mushroom bodies (MB). (**C**) Optic lobes (OL). (**D**) Medullae (ME). (**E**) Lobulae (LO). (**F**) Central complex (CX). (**G**) Other neuropils (OTH). Regression lines displayed with 95% confidence intervals. Pearson correlation coefficient (r) and p-value are given. Strong correlations ($r>0.40$) and significant correlations ($p<0.05$) are displayed in bold. (**H**) Linear correlations for the different neuropils (y-axis not given: differs for each neuropil). The grey dashed line indicates true isometric correlation (slope = 1).
(DOCX)

**S10 Fig. Correlation between relative volumes of neuropils and total brain volume (mm$^3$) for bumblebees (N = 77).** (**A**) Antennal lobes (AL). (**B**) Mushroom bodies (MB). (**C**) Optic lobes (OL). (**D**) Medullae (ME). (**E**) Lobulae (LO). (**F**) Central complex (CX). (**G**) Other neuropils (OTH). Regression lines displayed with 95% confidence intervals. Pearson correlation coefficient (r) and p-value are given. Strong correlations ($r>0.40$) and significant correlations ($p<0.05$) are displayed in bold. (**H**) Linear correlations for the different neuropils relative volume (y-axis not given: differs for each neuropil). The grey dashed line indicates true isometric correlation (slope = 1).
(DOCX)

**S11 Fig. Percentage of individuals per asymmetry categories for honey bees (*purple*) and bumblebees (*blue*).** AL and OL (N = 110 honey bees, N = 77 bumblebees), MB and AL and MB and OL (N = 59 honey bees, N = 36 bumblebees).
(DOCX)

**S1 Table. Average Dice scores of semi-automatic and automatic segmentation results of bumblebee (row 2) and honey bee brains (rows 3–12).** Outliers were automatically removed (see "Methods"). Last column: amount of manual correction required (Error in % of Dice score). Brain areas are labelled using the same abbreviations as in Fig 2. For some of the performance tests of the automatic segmentation (*yellow*), the 84 three-dimensional honey bee test images were split into 30 three-dimensional validation images and 54 three-dimensional test images (see "Methods"). Additionally, for honey bees that were initially scanned upside down, the 3D image data was flipped along the z-axis (yellow). It is important to note that all image data utilised in the study is three-dimensional.
(DOCX)

**S2 Table. Effect of varying hyperparameters on segmentation accuracy.** The reference score is based on Biomedisa's standard configuration as used in the manuscript. Highlighted Dice scores indicate that the average Dice score is statistically significantly smaller than the average Dice score of the reference sample (t-test, $p < 0.05$). Network size is represented by the number of filters up to the deepest block of the encoder part. The 110 three-dimensional honey bee images were divided into 26 training images, 30 three-dimensional validation images, and 54 three-dimensional test images. Similarly, the 77 three-dimensional bumblebee images were split into 13 three-dimensional training images, 20 three-dimensional validation images, and 24 three-dimensional test images. The final evaluation is conducted on the respective test images.
(DOCX)

**S3 Table. Correlation between absolute neuropil volumes (bottom left) and between relative neuropil volumes (top right) for honey bees (N = 110).** Pearson correlation coefficient and p-value are given. Strong correlations ($r > 0.40$) and significant correlations ($p < 0.05$) are displayed in bold. Brain areas are labelled using the same abbreviations as in Fig 2.
(DOCX)

**S4 Table. Inter-individual variability of total brain and neuropil volumes (%) within honey bee hives.** Percentage of volume variation (($\text{Max} - \text{Min}) \times 100 / \text{Max}$) was calculated for total brain and all neuropils for each hive. Brain areas are labelled using the same abbreviations as in Fig 2.
(DOCX)

**S5 Table. Total brain and neuropil volumes of bumblebees (N = 77).** Mean (± standard deviation), minimal and maximal volumes (mm$^3$). Global percentage of volume variation and inter-individual variability of brain and neuropils volume (%) within colonies (N = 77 bumblebees). Information is also given for the left and right sides of paired neuropils (i.e. AL, MB, OL, ME, LO). F-test, following LMMs, tests the significance of the fixed variable "colony", and results are displayed in bold when significant. Brain areas are labelled using the same abbreviations as in Fig 2.
(DOCX)

**S6 Table. Correlation between absolute neuropil volumes (bottom left) and between relative neuropil volumes (top right) for bumblebees (N = 77).** Pearson correlation coefficient and p-value are given. Strong correlations ($r > 0.40$) and significant correlations ($p < 0.05$) are displayed in bold. Brain areas are labelled using the same abbreviations as in Fig 2.
(DOCX)

## Author Contributions

**Conceptualization:** Philipp D. Lösel, Coline Monchanin, Renaud Lebrun, Jean-Marc Devaud, Vincent Heuveline, Mathieu Lihoreau.

**Data curation:** Philipp D. Lösel, Coline Monchanin.

**Formal analysis:** Philipp D. Lösel, Coline Monchanin.

**Funding acquisition:** Philipp D. Lösel, Vincent Heuveline, Mathieu Lihoreau.

**Investigation:** Philipp D. Lösel, Coline Monchanin.

**Methodology:** Philipp D. Lösel, Coline Monchanin.

**Project administration:** Philipp D. Lösel, Vincent Heuveline, Mathieu Lihoreau.

**Resources:** Philipp D. Lösel, Coline Monchanin, Vincent Heuveline.

**Software:** Philipp D. Lösel, Coline Monchanin, Alejandra Jayme, Jacob J. Relle.

**Supervision:** Vincent Heuveline, Mathieu Lihoreau.

**Validation:** Philipp D. Lösel.

**Visualization:** Philipp D. Lösel, Coline Monchanin.

**Writing – original draft:** Philipp D. Lösel, Coline Monchanin.

**Writing – review & editing:** Philipp D. Lösel, Coline Monchanin, Renaud Lebrun, Alejandra Jayme, Jacob J. Relle, Jean-Marc Devaud, Mathieu Lihoreau.

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
