## [Decision Letter · Decision Letter 0]

2 May 2023

Dear Dr. Lösel,

Thank you very much for submitting your manuscript "Natural variability in bee brain size and symmetry revealed by micro-CT imaging and deep learning" for consideration at PLOS Computational Biology.

As with all papers reviewed by the journal, your manuscript was reviewed by members of the editorial board and by several independent reviewers. In light of the reviews (below this email), we would like to invite the resubmission of a significantly-revised version that takes into account the reviewers' comments.

We cannot make any decision about publication until we have seen the revised manuscript and your response to the reviewers' comments. Your revised manuscript is also likely to be sent to reviewers for further evaluation.

Sincerely,

Barbara Webb

Academic Editor

PLOS Computational Biology

James O'Dwyer

Section Editor

PLOS Computational Biology

Reviewer's Responses to Questions

**Comments to the Authors:**

Reviewer #1: The Manuscript “Natural variability in bee brain size and symmetry revealed by micro-CT imaging and deep learning” by Lösel et.al. is investigating the variations in total brain size of bees. To do so, the authors used semi-automated and fully automated methods to segment micro-CT scans of different bee specimens. Finally, the authors performed statistical analysis of the volumes of the bee brains and their inner parts.

It is a nice work which has various merits; however, the manuscript has some important issues that need to be revised before it can be considered for publication:

In line 213, the authors claim that the dataset of their reference 29 is smaller though it is bigger. In Gronenberg -Couvillon publication the sample size of honeybees is 121.

The authors performed statistical analysis using 110 individuals of only one species of honeybees and 77 individuals of one species of bumblebees. Thus, it is a long stretch to generalize the results for all bee species, as there is a big variation between them. Therefore, I would suggest the authors to reconsider their title and be somewhat more precise/moderate.

It is unclear whether the authors considered the age and the body size of the individuals as a factor during their statistical analysis. They mention that “… workers vary little in body size” but did they compare the body sizes of their dataset? Was the head size, eyes, or antenna size measured and compared with the associated neuropils? Also, was the age of all individuals taken as a parameter in the linear mixed model? Finally, I would expect logarithmic transformation of the comparative data, as ecological data are usually skewed to the right.

As we can see in table S3, the number of bees from each hive was quite different. We can see that even within this small size of hive sample, the variability is not trivial, especially for OTH. As the number of individuals in each hive sample is getting bigger so does the variability in brain volume. Thus, how can the authors generalize their results for all honeybees? How did the authors choose the fixed and random factors? Were the specific hives representatives of different features in honeybees? Was the population treated as numerical factor and if not, what was its threshold?

The authors describe the statistical analysis of the variations within colonies but not the analysis between colonies.

It is not very clear when the authors talk about sets of images, 3D images, 2D images, or scans; e.g., in line 134 they talk about different numbers of images, in line 141 about scans, in line 163 about 3D images. Even though they are all the same, it is better to keep one description for the readers’ benefit.

The authors mention in line 178 that their automatic segmentation is highly accurate, which is too subjective a statement. 12.5% of an error greater than 4% is higher than the 9.4% with very few errors in bumblebees. That, I believe, comes as a result of an overfitted network due to the lack of training data. I would suggest the authors used transfer learning algorithms to retain the weights from honeybees and refine them for the bumblebees. In that way their network will be more generalized.

It is unclear why the authors chose 200 training epochs in such a small sample. Also, why the variation set is larger than the training set? How did the authors choose to split their data?

It is common to choose the values of hyperparameters based on the accuracy and performance of the network during training and validation. Can the authors provide such data? How was the loss changing during training? An ROC curve is essential to check the overfitting or underfitting of the network.

In S1 Fig. it is unclear whether the network overpredicted the areas or mislabelled them. It would be much better if the authors provided the ground truth image too.

Based on all these points, I think the manuscript needs to be significantly revised to warrant publication in PLOS Computational Biology.

Reviewer #2: The authors report on automated brain image analysis that facilitates large-scale analysis of neuropils and their variability in brain architecture in social Hymenoptera. By choosing social insects that exhibit elaborated behavior, such as honey- and bumble bees the authors report on lateralization of brain structures thus partly confirming partly previous reports on lateralization in bumble bees. An automated AI guided approach, as reported here using the open-source platform Biomedisa is very much needed in comparative brain analysis in evolutionary biology. The usage of the Biomedisa’s smart interpolation is already in public use. The novelty of the current manuscript is, as to my knowledge, its extension of the AI guided automation for analysis a large number of specimens.

The contrast in the CT scans is of sufficient quality to provide a clear separation of soma layers vs. neuropil, and neuropil vs neuropil, though the resolution (5.4 isotropic voxel size) is quite low compared to high-resolution CT or standard confocal imaging. (See my comments below).

In general, I find the approach promising from its technical side, and therefore potentially recommend their publication in PLOS Biology.

The results should be discussed more profound and compared to other approaches in its efficiency to facilitate segmentation of brain neuropils (see below). I regard this important knowledge for researchers prior to advance to large-scale scanning of specimen, i.e., there is a minimum of about 20 samples, that researcher have to sample for their experiments as a training set for the AI .

-Is the CT methods as efficient as a neuropil- specific marker compared to antibody based synapsin or fluorescent marker such as Lucifer yellow.

For example

- are other histological procedures (neuropil marker Synapsin, Lucifer Yellow) and scanning times less time consuming and giving better contrast (thus decreasing the number of training images for optimal segmentation) and might be better suited for AI guided segmentation?

- Muratore (2022) using synapsin stained brain tissue (Muratore IB, Fandozzi EM, Traniello JFA. Behavioral performance and division of labor influence brain mosaicism in the leafcutter ant Atta cephalotes. J Comp Physiol A. 2022;208(2):325-44.) The authors might want to cite and discuss their segmentation strategy. Synapsin antibodies are most commonly used neuropil marker in insects, giving high contrast. It would be interesting to see of how efficient the AI guided segmentation tools of BioMedisa is compared to the Muratore study.

- A simple, fast histology was employed in honeybees using Lucifer Yellow as a neuropil marker by Rybak 2010 (Rybak J, Kuss A, Lamecker H, Zachow S, Hege HC, Lienhard M, et al. The Digital Bee Brain: Integrating and Managing Neurons in a Common 3D Reference System. Front Syst Neurosci. 2010;4:30.). An statistical shape model composed out of 20 training images was then used for automated segmentation. In this respect: In your approach - does the pre-segmentation need to be done for each specimen over and over again. Or does the AI builds a a memory of relative position and /or shape of neuropiles during the training procedures?

- what exactly means: cropping of the data during segmentation ?

- In your segmented data sets the color of paired neuropils appear the same color for left and brain hemispheres. How does you, the AI indeed distinguish left/right ?

- in this respect: Automatic segmentation: how does the AI knows the identity of the neuropil 3D objects, or was the naming done manually?

- Comparing the volume data of this CT study (Table 1) with Brandt (2005: Table 2) that uses synapsin histology but the same definition of neuropils (except that for the CB the PB and Noduli of the CX were omitted in Brandt (2005): the absolute volume of all neuropils is app. 70-75 % smaller compared to the CT volume of this study, with exception of the MB neuropil this 58 % smaller. The reason for that might the histological procedure (more shrinkage in synapsin histology. In relative numbers (% neuropil/total brain) the MB differ by 5 % (CT = 25 %, confocal synapsin = 20 %).

- Although it is not in the focus of your study, I suggest to add a column with the relative volumes given as percentage in Table 1. That will allow an easier comparison with previous and future studies. I suggest to mark and separate the OL in Table 1 / Figure 5 as it is the sum of ME and LO.

- Brain = AL +MB +ME +LO +CX +OTH ?

- In Figure 5 n=59 for MB ?

- All raw data of all specimen should be provided in a supplementary excel tables.

- Though the quality of automated segmentation is good, I am puzzled by the high amount of fused MBs in the samples, about the half of the Apis specimen. What could the reason? The most distal medial lobes tend to merge into each other. May be an sub-script can be programmed to create a automated separation?

- A clear separation from connective tracts such as the lobula-to-optic tubercle tract seems to be difficult in CT imaging and these tracts are partly reconstructed (as seen in the reconstruction of Figure 2E). Where this tract belongs to? the lobula or the OHT (protocerebral lobe)?

- Optical tubercle: in your CT scans the optic tubercle was very well distinguishable. Why not take the volume data in your list. The optic tubercle is one of the main outputs of the OL to the protocerebral lobe.

- line 31 : 110 honeybees and 77 bumble bees ?

- are all species foragers as in the honeybees ? if so variations might be due to foraging experience, too ?

- line 73: bees and some and ants beetles are model species

- line 139: in the remaining 84 CTs you did not perform manual corrections on selected pre-labeled slice ? Or, how are these checked for quality ? In bumble bee, see below, line 152 you did not perform any quality check ?

- line 174-176: it would be useful to give the average time / one brain for segmentation times

- also it should be mentioned that the automated, and linear segmentation of Amira segment Tools, including water-shed, considerably reduce the ‘pure’ time of manual segment each slice.

Depending on the contrast, simple threshold segmentation sometimes is sufficient to segment neuropils very fast… In other words: what is ‘purely’ manual segmentation, indeed ?

- line 232: study by Gronenberg(2010) : lightmicroscopy but not electronmicroscopy !! They do compare 67 european vs 54 africanized honeybees.

- line 233: relative volume: are your data different from Gronenberg (2010) ? discussion

- line 241: you already reported relative volume in line 233 ? your statements are confusing here.

- line 762 ASSD?

- line 765 un-copped ?

Figure1

- on the left side the time is noted for one specimen, but on the right side an assembly of brains are shown. Times for manual correction should be noted accordingly.

- Are the manual corrections always done on selected slices? or the whole volumen ?

- Does the ‘smart Biomedisa interpolation’ include more sophisticated algorithm than e.g. the water-shed interpolation of Amira ?

Figure 3

- line 765 correct auto-copping

Figure 4 revise

- the figure is difficult to read and confusing: Haddad (35) deep blue and then again purple ?

- volume in mm^3 for the y-axis ? provide the raw data

- the black circles (CT-scan)I can’t see it in the graph ?

Figure 6

- what is LMM

- are inter individual variations within each have shown here ? and there is not variations between the two hives ? Indicate and explain.

Figure 7

- mm^3 for the y-axis ?

- why are the mean from other CT studies so different ? Are they ?

- line 782-783 I do not understand the sentence. In Brandt (2005) and Haddad the volume were measured in order to eliminate all variations that are due to segmentation. In contrast: other studies were done to measure natural variability ….

- The authors provide volume in cubic millimeter. In insects volume measures are mostly in cubic-micrometer (scientific format). I took the time to convert and compare the data from this study with the data from Brandt (2005). To fascilitate with most other insect studies can you provide an excel tables with your numbers and properly convert ?

The mismatch might be due to a different definition of neuropil borders and / or a different contrast in CT versus confocal synapsin staining. For example, the tracts between medulla and lobula are difficult to differentiate.

A final comment: How much of the reported variability is due to natural variability, and how much contribute the technical and segmentation procedures to the variance of your data?

Reviewer #3: In this paper, the authors present a deep learning method for analysis of 3D volumes. The method is applied to volumes of bee brains to demonstrate that it can effectively identify and segment the volumes of different brain areas. The authors then perform some comparative analyses of the data and make some inferences about how this relates to inter-individual and inter-species differences.

In general, I think that this method seems like an effective way to segment insect brain volumes and the paper is well-written and clear. I do think that the paper is sufficiently rigorous and of general interest to warrant publication. I have some relatively minor comments however that I think should be addressed to improve clarity.

Line 90: please provide the species here.

Line 134 (and, for example, line 455): it is confusing to call the image stacks for 3D volumes 'images' and this could lead to misunderstandings. Throughout the text, please be clear when referring to 'volumes' and when referring to images (i.e. 2D images).

Line 285: I do not understand how this statement follows logically from the previous one. I do not think that any evidence is presented to suggest that there are adaptive levels of brain size variation and I therefore think that this sentence should be removed. At no point were divisions of labour taken into account. All honey bees were foragers and the bumblebees were selected randomly and therefore their roles were unknown. While there is some evidence of a trend for smaller bumblebees to take on tasks in the hive, while larger ones are foragers, this is not at all clear and differs from hive to hive.

Line 346: As with the comment above, there is no evidence provided in this paper that supports this statment. The paper cited–55–certainly is not appropriate here as it does not attempt at all to relate brain asymmetries with cognitive variability. The first part of this sentence should be removed.

Line 361: As with my comments above, there is no evidence provided in the paper that supports the statement that there is 'selection for variability supporting adaptive division of labour' and this part of the sentence should be removed.

Figure 5F: The y-axis does not show any numerical values apart from 0, please fix.

Figure 6: I do not find any information about the number of bees taken from each colony. Please provide this in the figure text.

Figure 7: Separating the data from Rother et al. and Smith et al. is problematic as Rother et al. used data from Smith et al. so it is just a subset of their data and not a new data set. This should be removed. Also, it is not clear why this data is not presented as for the honey bees in Figure 5. I think that this should be done.

Figure 8: it would be more informative if the correlations were made between the same individual rather than across the whole data set so a paired test should be used but this is not stated in the figure text.

**Have the authors made all data and (if applicable) computational code underlying the findings in their manuscript fully available?**

Reviewer #1: None

Reviewer #2: **No: **the raw volume data for individual 187 specimen are missing

Reviewer #3: Yes

PLOS authors have the option to publish the peer review history of their article (what does this mean?). If published, this will include your full peer review and any attached files.

Reviewer #1: No

Reviewer #2: **Yes: **Jürgen Rybak

Reviewer #3: No
---

## [Decision Letter · Decision Letter 1]

19 Sep 2023

Dear Dr. Lösel,

We are pleased to inform you that your manuscript 'Natural variability in bee brain size and symmetry revealed by micro-CT imaging and deep learning' has been provisionally accepted for publication in PLOS Computational Biology.

Best regards,

Barbara Webb

Academic Editor

PLOS Computational Biology

James O'Dwyer

Section Editor

PLOS Computational Biology

Reviewer's Responses to Questions

**Comments to the Authors:**

Reviewer #1: The review is uploaded as an attachment.

Reviewer #2: there are no further details in the manuscript to be revised

Reviewer #3: I thank the authors for considering my comments. I am satisfied with all changes and responses and think that the paper is now suitable for publication.

**Have the authors made all data and (if applicable) computational code underlying the findings in their manuscript fully available?**

Reviewer #1: Yes

Reviewer #2: Yes

Reviewer #3: Yes

PLOS authors have the option to publish the peer review history of their article (what does this mean?). If published, this will include your full peer review and any attached files.

Reviewer #1: No

Reviewer #2: **Yes: **Jürgen Rybak

Reviewer #3: No

---

## [Editor Report · Acceptance letter]

28 Sep 2023

PCOMPBIOL-D-23-00086R1 

Natural variability in bee brain size and symmetry revealed by micro-CT imaging and deep learning

Dear Dr Lösel,

I am pleased to inform you that your manuscript has been formally accepted for publication in PLOS Computational Biology. Your manuscript is now with our production department and you will be notified of the publication date in due course.

With kind regards,

Anita Estes
